# Neoadjuvant Afatinib for stage III *EGFR*-mutant non-small cell lung cancer: a phase II study

Dongliang Bian [1,10], Liangdong Sun[1,10], Junjie Hu[1,10], Liang Duan[1,10], Haoran Xia[1,10], Xinsheng Zhu[1], Fenghuan Sun[1], Lele Zhang [2], Huansha Yu[3], Yicheng Xiong[1], Zhida Huang[1,4], Deping Zhao[1], Nan Song[1], Jie Yang[1], Xiao Bao[5], Wei Wu[6], Jie Huang[7], Wenxin He[1,11] ✉, Yuming Zhu[1,11] ✉, Gening Jiang[1,11] ✉ & Peng Zhang [1,8,9,11] ✉

Afatinib, an irreversible ErbB-family blocker, could improve the survival of advanced epidermal growth factor receptor (*EGFR*)-mutant non-small cell lung cancer patients (NSCLCm+). This phase II trial (NCT04201756) aimed to assess the feasibility of neoadjuvant Afatinib treatment for stage III NSCLCm+. Forty-seven patients received neoadjuvant Afatinib treatment (40 mg daily). The primary endpoint was objective response rate (ORR). Secondary endpoints included pathological complete response (pCR) rate, pathological down-staging rate, margin-free resection (R0) rate, event-free survival, disease-free survival, progression-free survival, overall survival, treatment-related adverse events (TRAEs). The ORR was 70.2% (95% CI: 56.5% to 84.0%), meeting the pre-specified endpoint. The major pathological response (MPR), pCR, pathological downstaging, and R0 rates were 9.1%, 3.0%, 57.6%, and 87.9%, respectively. The median survivals were not reached. The most common TRAEs were diarrhea (78.7%) and rash (78.7%). Only three patients experienced grade 3/4 TRAEs. Biomarker analysis and tumor microenvironment dynamics by bulk RNA sequencing were included as predefined exploratory endpoints. CISH expression was a promising marker for Afatinib response (AUC = 0.918). In responders, compared to baseline samples, increasing T-cell- and B-cell-related features were observed in post-treatment tumor and lymph-node samples, respectively. Neoadjuvant Afatinib is feasible for stage III NSCLC+ patients and leads to dynamic changes in the tumor microenvironment.

Non-small cell lung cancer (NSCLC) accounts for 85% of lung cancer cases[1]. A higher proportion of NSCLC patients have epidermal growth factor receptor (*EGFR*) mutations, particularly among women, never smokers, East Asians, and those with adenocarcinoma[2,3]. The most common *EGFR*-mutant subtypes include exon 19 deletions (Ex19Del) and exon 21 codon p.Leu858Arg point mutation (L858R)[4,5]. *EGFR*

tyrosine kinase inhibitors (*EGFR*-TKIs) have been recommended as the first-line treatment for advanced *EGFR*-mutant NSCLC patients (NSCLCm+)[6].

More than 15% of patients diagnosed with NSCLC are in locally advanced stage[5]. As the stage of NSCLC advances from IIIA to IIIC (the 8th edition lung cancer TNM staging system), the 5-year overall

A full list of affiliations appears at the end of the paper. ✉e-mail: 0wenxinhe@tongji.edu.cn; ymzhu2005@aliyun.com; geningjiang@tongji.edu.cn; zhangpeng1121@tongji.edu.cn

survival (OS) rate decreases from 36% to 13%[1]. For locally advanced NSCLCm+, current clinical guidelines recommend neoadjuvant therapy followed by surgery and postoperative adjuvant platinum-based chemotherapy followed by *EGFR*-TKI therapy[7–10]. However, despite these treatment regimens, the prognosis for these patients remains poor. Recent clinical trials have evaluated the prognosis of neoadjuvant treatment followed by surgery for NSCLCm+ with stage III[11–14]. In a randomized phase II study (EMERGING-CTONG 1103), it was demonstrated that neoadjuvant Erlotinib compared to platinum-based chemotherapy significantly improved the progression-free survival (PFS) (21.5 months vs 11.4 months) and objective response rate (ORR) (54.1% vs 34.3%)[12]. Additionally, a single-arm, phase II study showed that neoadjuvant Gefitinib (ORR: 54.5%) followed by surgery is a feasible therapy regimen for operable NSCLCm+ in locally advanced stage[13].

Afatinib is a second-generation *EGFR*-TKI that effectively inhibits members of the ErbB-family (*EGFR*, *HER2*, and *ErbB4*) through irreversible binding[15–17]. In a randomized phase II study (LUX-Lung 7), Afatinib showed superior outcomes compared to Gefitinib in advanced NSCLCm+ patients (PFS: 11.0 months vs 10.9 months, HR: 0.73, 95% CI: 0.57–0.95, $p = 0.017$; ORR: 70% vs 56%, odds ratio (OR): 1.87, 95% CI: 1.18–2.99, $p = 0.00083$)[18]. A single-arm phase III study conducted in China further confirmed the feasibility of Afatinib as a therapy for Chinese patients with advanced NSCLCm+ (ORR: 59.1%, PFS: 11.4 months) and did not reveal any new safety concerns[19]. Consequently, Afatinib as a neoadjuvant treatment might offer advantages over first-generation EGFR-TKIs for locally advanced NSCLCm+. However, there are limited studies discussing the efficacy and safety of Afatinib followed by radical resection for these patients.

This work, the single-arm phase II clinical trial (TEAM-LungMate 004), aims to assess the efficacy and safety of neoadjuvant Afatinib for stage III NSCLCm+. The trial successfully achieves the pre-specified primary endpoint with an objective response rate (ORR) of 70.2%. Moreover, the study delves into investigating the molecular characteristics that are associated with the efficacy of Afatinib, as well as exploring the dynamic changes occurring in the tumor

microenvironment following Afatinib treatment through bulk RNA sequencing (RNA-seq) on baseline and post-treatment samples obtained from the trial participants.

## Results

### Baseline characteristics

Between July 2020 and February 2022, a total of 49 eligible NSCLCm+ in stage III were screened (Fig. 1). Among them, 47 participants were enrolled in this study as the intention-to-treat (ITT) population, as indicated in Table 1. The average age at diagnosis was 60.6 years (range, 33–78 years), and the majority of participants were female (26/47, 55.3%), never-smokers (36/47, 76.6%), and had an ECOG performance status of 0 (29/47, 61.7%). Among the enrolled participants, 28 (59.6%) had the Ex19Del mutation, while 16 (34.0%) had the L858R mutation (Supplementary Table 1). Additionally, 3 participants (6.4%) had uncommon *EGFR* mutations (Supplementary Table 2). The mean maximum tumor dimension (MTD) of all participants was 43.2 mm (range, 18–99 mm). Most participants were categorized as stage N2 (32/47, 68.1%). Among the enrolled participants, 26 (55.3%) were classified as stage IIIA, 18 (38.3%) as stage IIIB, and 3 (6.4%) as stage IIIC.

### Efficacy

The information of the efficacy of neoadjuvant Afatinib therapy (NAT) was listed in Table 2. Following NAT (mean cycles: 2.7, range, 1–9 cycles), 33 patients (70.2%) achieved a partial response (PR), 11 patients (23.4%) had stable disease (SD), and 2 patients (4.3%) experienced progressive disease (PD). Only one participant had to change the therapeutic regimen due to grade 4 diarrhea during the first NAT cycle and declined further evaluation at our center. Waterfall plots illustrating the responses are shown in Fig. 2a. The primary endpoint, ORR, was 70.2% (95% confidence interval (CI): 56.5% to 84.0%). Regarding secondary endpoints, the rate of margin-free resection (R0) among the 33 participants who underwent NAT followed by surgery was 87.9% (95% CI: 71.8–96.6%). The

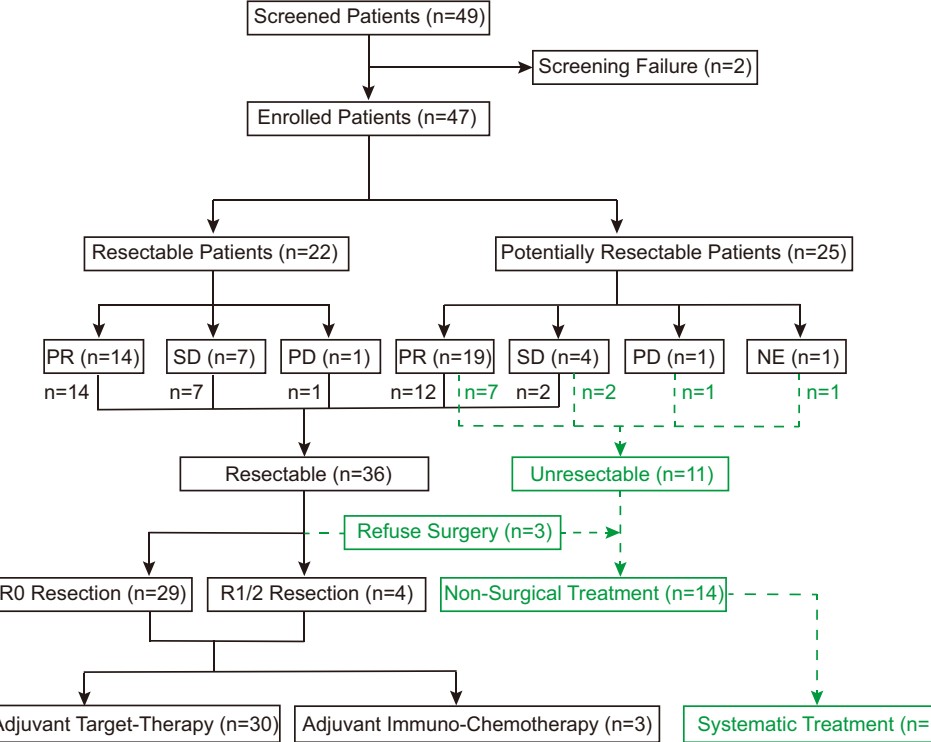

**Fig. 1 | The clinical trial (TEAM-LungMate 004) design and clinical efficacy.** NE: non-evaluation; PD: progressive disease; PR: partial response; R0 Resection: complete resection; R1/2 Resection: uncomplete resection; SD: stable disease.

**Table 1 | Baseline characteristics of patients received neoadjuvant Afatinib therapy (_n_ = 47)**

| Variables | No. of patient (%) |
|---|---|
| **Age, years (range)** | 60.6 (33–78) |
| **Gender** | |
| Male | 21 (44.7) |
| Female | 26 (55.3) |
| **ECOG performance status** | |
| 0 | 29 (61.7) |
| 1 | 18 (38.3) |
| **_EGFR_ mutation** | |
| Ex19Del | 28 (59.6) |
| L858R | 16 (34.0) |
| G719X | 2 (4.3) |
| S768I | 1 (2.1) |
| **Smoking status** | |
| Ever | 11 (23.4) |
| Never | 36 (76.6) |
| **MTD at diagnosis, mm (range)** | 43.2 (18–93) |
| **N Stage at diagnosis** | |
| 0 | 0 (0.0) |
| 1 | 5 (10.6) |
| 2 | 32 (68.1) |
| 3 | 10 (21.3) |
| **cTNM stage at diagnosis** | |
| IIIA | 26 (55.3) |
| IIIB | 18 (38.3) |
| IIIC | 3 (6.4) |
| **Operable evaluation at diagnosis** | |
| Resectable | 22 (46.8) |
| Potentially resectable | 25 (53.2) |

_MTD_ maximum tumor dimension.

**Table 2 | Evaluation of efficacy of neoadjuvant Afatinib (_n_ = 47)**

| Variables | No. of patient (%) |
|---|---|
| **Duration of NAT, cycle (range)** | 2.7 (1-9) |
| **Surgery performed** | 33 (70.2) |
| **Tumor response** | |
| PR | 33 (70.2) |
| SD | 11 (23.4) |
| PD | 2 (4.3) |
| No Re-evaluation | 1 (2.1) |
| **ORR (%)** | 70.2 |
| **DCR (%)** | 93.6 |
| **MTD regression, % (range)** | 36.0 (−28–78) |
| **cN downstaging** | 37 (78.7) |
| **ypN downstaging [#]** | 19 (57.6) |
| **ypTNM [#]** | |
| 0/I | 14 (42.4) |
| II | 5 (15.2) |
| III | 14 (42.4) |
| **Type of surgery [#]** | |
| Thoracotomy | 16 (48.5) |
| VATS | 17 (51.5) |
| **Surgical resection [#]** | |
| Lobectomy | 28 (84.8) |
| Bi-Lobectomy | 5 (15.2) |
| **Resection [#]** | |
| R0 | 29 (87.9) |
| R1 | 1 (3.0) |
| R2 | 3 (9.1) |
| **Residual tumor cell, % (range) [#]** | 57.3 (0-95) |
| **Residual tumor cell ≤ 60% [#]** | 17 (51.5) |
| **Pathologic regression [#]** | |
| pCR | 1 (3.0) |
| MPR | 3 (9.1) |
| Non-MPR | 30 (90.9) |
| **Mean operation time (hour)** | 2.4 |
| **Mean blood loss (mL)** | 100 |
| **Mean hospital stay postoperatively, day (range)** | 6.6 |
| **Mean chest drainage time, day (range)** | 12.5 |

#: Patients received neoadjuvant therapy followed by surgery.
_DCR_ disease control rate, _MPR_ major pathological response, _MTD_ maximum tumor dimension, _NAT_ neoadjuvant Afatinib treatment, _ORR_ objective response rate, _pCR_ pathological complete response, _PD_ progressive disease, _PR_ partial response, _SD_ stable disease.

major pathologic response (MPR) was evaluated in these 33 samples (Fig. 2b). Among the three participants who achieved MPR (MPR rate: 9.1%, 95% CI: 1.9% to 24.3%), only one participant achieved pathologic complete response (pCR). Furthermore, the mean percentage of residual tumor cells in the primary tumor after surgery was 57% (range, 0% to 95%). The average duration of postoperative hospital stay was 6.6 days, and chest drainage was maintained for an average of 12.5 days. During surgery, the mean operation time was 2.4 h, with mean blood loss of 100 mL. All 33 participants who underwent surgical treatment received adjuvant treatment postoperatively, including target-therapy (_n_ = 30) and immunochemotherapy (_n_ = 3). Adjuvant immunochemotherapy for the 3 participants was administered based on the following reasons: 1) extremely low EGFR mutant abundance in the resected tumor tissue; 2) diagnosis of double primary tumors harboring EGFR and KRAS mutations, respectively, postoperatively; 3) primary Afatinib resistance.

## Survival analysis

At the cut-off date of May 1st 2023, the median duration of follow-up for all participants was 24.0 months (interquartile range (IQR), 20.0-30.0 months). The median event-free survival (EFS) for all participants was not reached (NR) (Fig. 3a). No significant differences in EFS were observed among participants in different gender, mutant subtype, and diagnostic TNM stage subgroups (_p_ = 0.325, 0.421, 0.944, respectively) (Fig. 3b–d). Based on the evaluation of the time-dependent Cox model, a significant improvement in EFS was

observed for participants who experienced a proportionate reduction in sum of lesion diameter (SLD) on radiological evaluation (HR: 0.954, 95% CI: 0.924–0.985, _p_ = 0.004). However, no clear trend in EFS was observed in relation to pathological response and surgical treatment (Supplementary Table 3). As of the last follow-up, a total of 18 participants experienced events, with 11 of them having postoperative recurrence. Among the recurrent sites, the proportion of participants with metastasis in the central nervous system (CNS) was the highest at 45.5% (5/11).

## Stratified analysis

The baseline information for different _EGFR_-mutant subtypes and resectable subgroups was manifested in Supplementary Table 1.

As shown in Supplementary Table 4, the participants with Ex19Del mutation had a lower ORR than those with L858R mutation (64.3% vs 81.3%, _p_ = 0.235). Additionally, Ex19Del participants

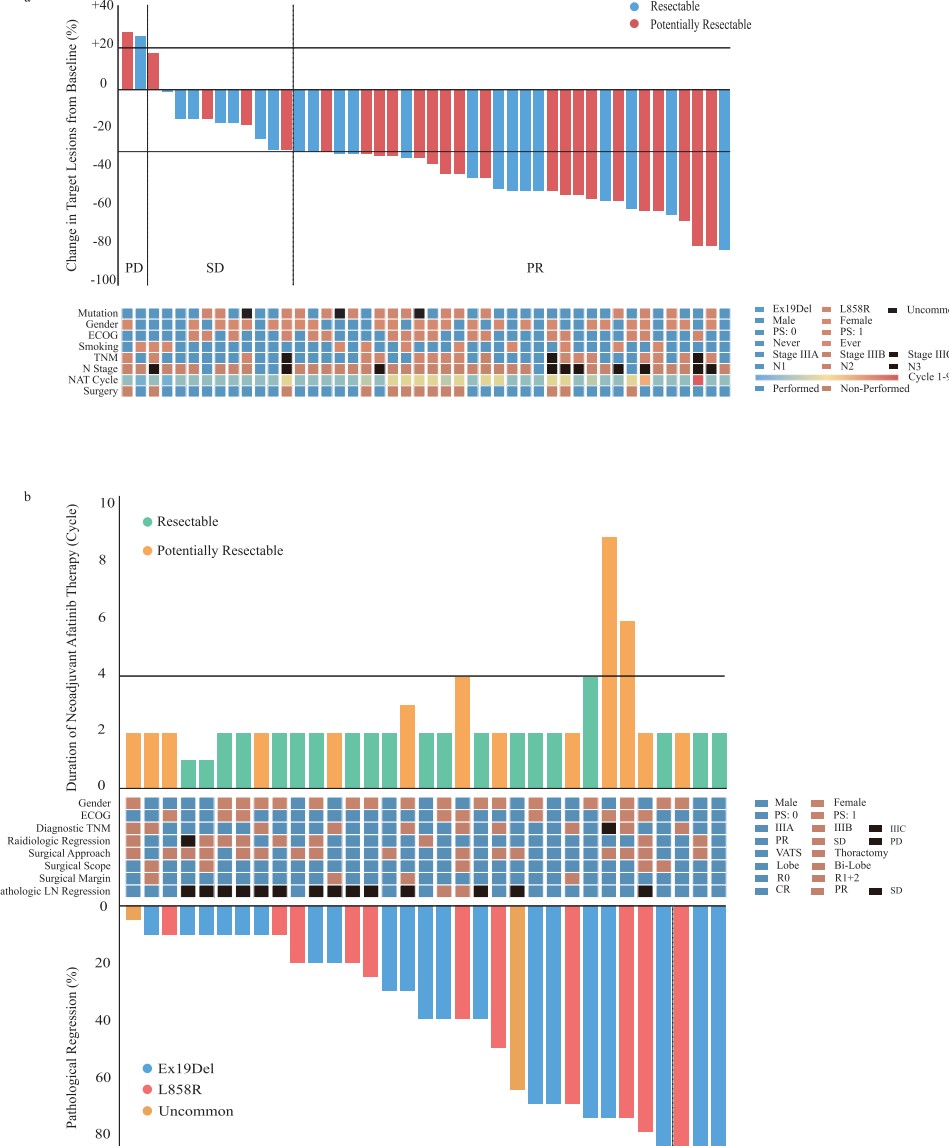

**Fig. 2 | Radiological and pathological outcomes of participants. a** Waterfall plots of radiological regression of target lesions for patients after neoadjuvant Afatinib treatment (*n* = 46). **b** Evaluation the relationship between the duration of neoadjuvant Afatinib therapy and pathological regression of patients after surgery

(*n* = 33). Source data are provided as a Source Data file. LN: lymph node; MPR: major pathologic response; NAT: neoadjuvant Afatinib treatment; PD: progressive disease; PR: partial response; R0 Resection: complete resection; R1/2 Resection: uncomplete resection; SD: stable disease; VATS: video-assisted thoracic surgery.

showed a significantly lower radiological regression of target lesions compared to L858R participants (35.3% vs 38.9%, *p* = 0.046). Two Ex19Del participants experienced PD after NAT due to primary Afatinib resistance, and two Ex19Del participants achieved SD after NAT due to extremely low mutant abundance. These factors may explain the discrepancy in ORR between these two subgroups. However, no significant differences were observed between Ex19Del and L858R participants in terms of R0 and MPR (Table 3).

As shown in Supplementary Table 5, resectable participants had a lower ORR after NAT compared to potentially resectable participants (63.6% vs 76.0%, *p* = 0.355), which may be attributed to the significantly shorter duration of NAT in the resectable group (cycles: 3.2 vs 2.2, *p* = 0.016). Among the 22 resectable patients, except for two participants who declined surgical treatment, the remaining 20 participants received complete resection (surgery: 90.9%, R0: 100.0%, MPR: 10.0%). Among the 25 potentially resectable participants, 13 participants received surgery after NAT, and 9

of them underwent radical resection (surgery: 52.0%, R0: 69.2%, MPR: 7.7%). This indicated that neoadjuvant Afatinib therapy may provide surgical opportunities for inoperable stage III NSCLCm+. Moreover, prolonging the duration of Afatinib for local advanced NSCLCm+ preoperatively could provide benefit for the ORR, but not the MPR.

## Safety

Treatment-related adverse events (TRAEs) were listed in Table 3. All participants (100%) experienced TRAEs, but no grade 5 TRAEs were observed. Grade 3/4 TRAEs occurred in 3 patients (6.4%), including diarrhea, interstitial pneumonia, and hepatic impairment, respectively. Discontinuation of Afatinib was observed in only one participant due to grade 4 diarrhea. The most common TRAEs during NAT were diarrhea (78.7%), rash (78.7%), stomatitis and oral ulcers (68.1%), paronychia (51.1%), anorexia and nausea (25.5%), fatigue (23.4%), and pruritus (21.3%).

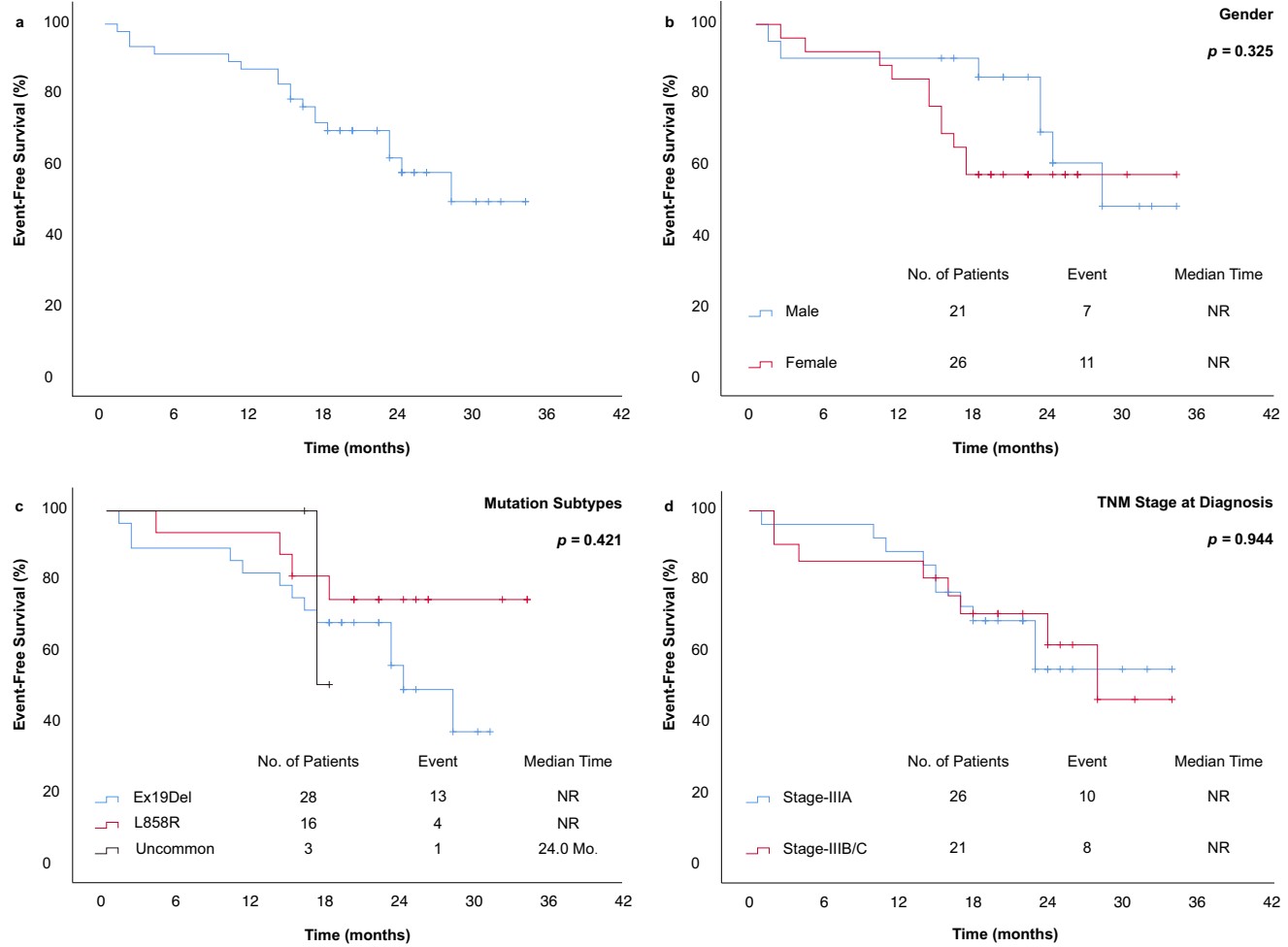

**Fig. 3 | Kaplan-Meier estimates of patients EFS for the ITT population.** The EFS of (**a**) total ITT (*n* = 47), (**b**) Males and Females, (**c**) different mutation subtypes, (**d**) different TNM stage at diagnosis. *P*-value was calculated using stratified log-rank test, with *p* < 0.05 considered statistically significant. Source data are provided as a Source Data file. EFS: event-free survival; ITT: intention-to-treat; NR: not reached.

Surgery-related complications occurred in four participants, including prolonged air leakage (3/33, 9.1%) and bronchopleural fistula (1/33, 3.0%). However, all the 3 participants recovered from complications through non-surgical procedures. No surgery-related death occurred within 90 days postoperatively.

**Exploratory endpoint**

In order to identify reliable biomarkers for responders, bulk RNA-seq and immunohistochemical (IHC) score were conducted on the primary tumor. Additionally, to explore the molecular characteristics of tumor microenvironment (TME), RNA-seq was performed on the primary tumor and lymph-node (LN) samples at pre-treatment (baseline) and post-treatment phases (Fig. 4a).

**Differences in molecular characteristics at baseline.** Based on baseline tumor samples, we observed that Cytokine-Inducible SH2-Containing Protein (CISH) exhibited higher expression in responsive tumor samples compared to non-responsive samples at baseline (10 responsive samples vs 7 non-responsive samples, *p* = 0.007) (Fig. 4b). A similar trend was confirmed in the CISH-IHC score (6 responsive samples vs 6 non-responsive samples, *p* = 0.100) (Fig. 4f). CISH expression at baseline performed well in predicting positive response for Afatinib, as evidenced by RNA-seq and IHC score (area under the dose-response curve (AUC) = 0.918 and 0.792, respectively) (Figs. 4c, g). Additionally, the participants in this trial with higher CISH expression in RNA-seq exhibited a favorable tendency in terms of OS and EFS (*p* = 0.140 and 0.057, respectively) (Fig. 4d, e), as well as in CISH-IHC score (*p* = 0.130 and 0.110, respectively) (Fig. 4h, i). The representative IHC images

### Table 3 | Adverse events in neoadjuvant therapy and surgery

| Type of event | No. of patients | |
|---|---|---|
| | All grades (%) | Grade ≥ 3 (%) |
| **Adverse event (*n* = 47)** | 47 (100.0) | 3 (6.4) |
| Diarrhea | 37 (78.7) | 1 (2.1) |
| Rash | 37 (78.7) | 0 (0.0) |
| Stomatitis | 32 (68.1) | 0 (0.0) |
| Paronychia | 24 (51.1) | 0 (0.0) |
| Anorexia and nausea | 12 (25.5) | 0 (0.0) |
| Fatigue | 11 (23.4) | 0 (0.0) |
| Pruritus | 10 (21.3) | 0 (0.0) |
| Dry skin | 4 (8.5) | 0 (0.0) |
| Interstitial pneumonia | 3 (6.4) | 1 (2.1) |
| Elevated ALT | 3 (6.4) | 1 (2.1) |
| **Surgical-related complications (*n* = 33)** | | |
| Prolonged air leakage | 3 (9.1) | 0 (0.0) |
| Bronchopleural fistula | 1 (3.0) | 0 (0.0) |

*ALT* alanine aminotransferase.

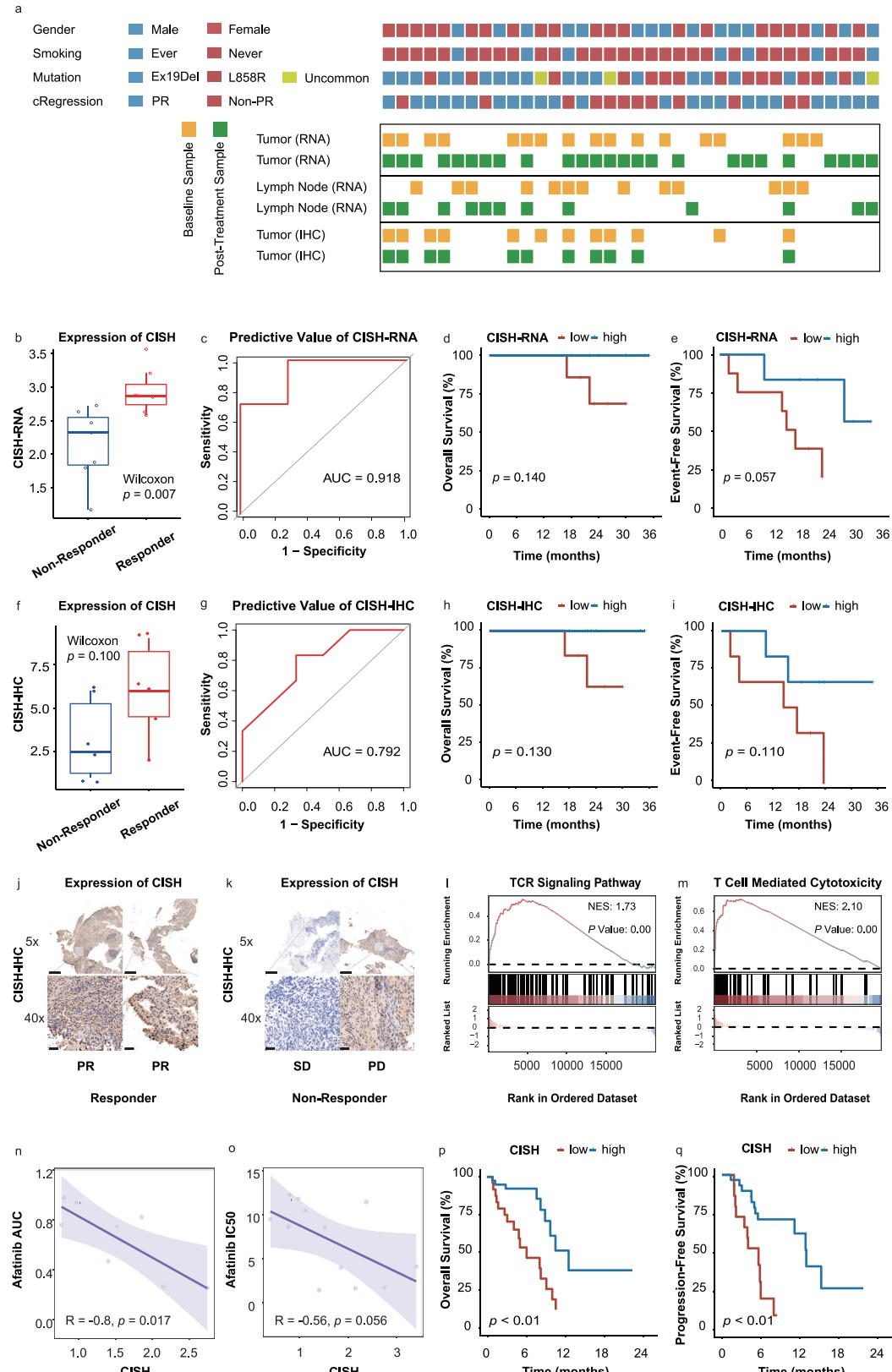

for CISH in each responsive group were shown (Fig. 4j, k). Higher CISH RNA-seq expression group showed enrichment in the TCR signaling pathway and T-cell-mediated cytotoxicity pathway (Fig. 4l, m), which might contribute to better therapeutic efficacy of *EGFR*-TKI. Moreover, in lung adenocarcinoma (LUAD) cell lines with *EGFR* mutation from DepMap, CISH expression exhibited a negative correlation with the AUC and half the maximal inhibitory concentration (IC50) of Afatinib, suggesting that the tumor cells with higher expression of CISH were more sensitive to Afatinib (Fig. 4n, o). Similarly, in the TCGA-LUAD cohort with *EGFR* mutations, patients with higher CISH expression had significantly better OS and PFS (both *p* < 0.01) (Fig. 4p, q).

**Fig. 4 | The biomarker of therapeutic efficacy in pre-treatment tumor samples.**
**a** RNA-seq and IHC for baseline and post-treatment samples. **b** CISH was highly expressed in baseline tumor samples in RNA-seq expression from tumor responders (*n* = 7) compared with non-responders (*n* = 7). **c** CISH RNA-seq expression in baseline samples could predict TKI efficiency (*n* = 14). **d** The difference in OS between high (*n* = 6) and low (*n* = 8) CISH RNA-seq expression. **e** The difference in EFS between high (*n* = 4) and low (*n* = 10) CISH RNA-seq expression. **f** CISH was highly expressed in baseline tumor samples in IHC expression from tumor responders (*n* = 6) compared with non-responders (*n* = 6). **g** CISH IHC expression in baseline samples could predict TKI efficiency (*n* = 12). **h** The difference in OS between high (*n* = 6) and low (*n* = 6) CISH IHC expression. **i** The difference in EFS between high (*n* = 6) and low (*n* = 6) CISH IHC expression. Representative IHC images (5X and 40X) for CISH in responder (**j**) and non-responder (**k**) tumors. Scare bar, 400μm for 5X images and 40μm for 40X images. Significant enrichment for TCR signaling pathway (**l**) and T cell mediated cytotoxicity pathway (**m**) in high

CISH RNA-seq expression group (*n* = 7). CISH expression was associated with Afatinib efficacy in DepMap LUAD cell-lines with *EGFR* mutations in (**n**) PRISM secondary screen (*n* = 8) and (**o**) CTRP database (*n* = 12), the regression line is blue, and the shading indicates the 95% CI. **p** The difference in OS between high and low CISH expression in TCGA-LUAD with *EGFR* mutations (*p* = 0.00022) (*n* = 59). **q** The difference in PFS between high and low CISH expression in TCGA-LUAD with *EGFR* mutations (*p* = 0.00078) (*n* = 50). Centers, boxes, whiskers, and dots indicate medians, quantiles, minima/maxima, and outliers, respectively in (**b**) and (**f**). Two-sided Wilcoxon rank sum test was used for comparison in **b** and **f**. Statistical comparisons in **d**, **e**, **h**, **i**, **p** and **q** were used by log-rank test. Source data are provided as a Source Data file. PR: partial response; SD: stable disease; PD: progressive disease; AUC: area under curve; IHC: immunohistochemistry; IC50: half maximal inhibitory concentration; TCR: T-cell receptor; CISH: Cytokine-Inducible SH2-Containing Protein.

**EMT was activated after Afatinib treatment in non-responders.** In the non-responsive tumors, analysis of longitudinal samples identified 1850 differentially expressed genes between baseline and post-treatment phases (Supplementary Fig. 1a). Gene set enrichment analysis (GSEA) revealed significant enrichment of the "estrogen response late" and "epithelial-mesenchymal transition" (EMT) pathways in post-treatment tumor samples (Supplementary Fig. 1b). Furthermore, a decreasing tendency in NK cells and an increasing tendency in myeloid dendritic cells were observed in post-treatment tumor samples (Supplementary Fig. 1c). Additionally, the expression of VEGF-B was found to be increased in post-treatment tumor samples compared to the baseline samples (Supplementary Fig. 1d). Regarding the LNs from non-responders, 3614 differentially expressed genes between baseline and post-treatment phase were identified in longitudinal LN samples (Supplementary Fig. 1e). GSEA analysis also indicated significant enrichment of the EMT pathway in LN samples after treatment (Supplementary Fig. 1f).

**Anti-tumor immunity was activated after Afatinib treatment.** The expression of programmed cell death 1 ligand 1 (PD-L1) was assessed by IHC score in 12 and 11 primary tumor samples before and after neoadjuvant Afatinib treatment in this trial, respectively. At baseline, 10 tumors (83.3%) were negative for PD-L1 expression, while 2 tumors (16.7%) showed positive expression. After NAT, 7 participants (7/11, 63.6%) exhibited positive PD-L1 expression in their primary tumor, including one participant with high expression. Analysis of paired samples before and after Afatinib treatment revealed elevated PD-L1 expression after treatment in 3 responders and 2 non-responders (Supplementary Fig. 2a, b). Furthermore, compared to non-responsive tumors, responsive tumor samples exhibited higher PD-L1 expression at baseline as determined by IHC score (*p* = 0.170) (Supplementary Fig. 2c). In addition, a significant increase in PD-L1 expression was observed in post-treatment tumor samples compared to baseline samples by IHC score (*p* = 0.013) (Supplementary Fig. 2d), while no such difference was observed at the transcriptome level (*p* = 0.800) (Supplementary Fig. 2e).

In longitudinal tumor samples from responders, a total of 1681 genes were found to be differentially expressed between baseline and post-treatment samples. After Afatinib treatment, 721 genes showed down-regulated expression, while 960 genes exhibited up-regulated expression (Fig. 5a). GSEA analysis identified that the "T cell receptor signaling" pathway was significantly enriched, indicating T-cell activation after Afatinib treatment. Consistently, the enrichment of "Glycolysis" was specifically down-regulated in responsive tumors after therapy (Fig. 5b). It is reported that the lactate from tumor glycolysis could promote an immunosuppressive environment by impairing the anti-tumor effect of CD8 + T-cells, inducing PD-1 expression in regulatory T-cells, and recruiting myeloid-derived suppressor cells (MDSCs)[20,21]. After treatment, an increasing tendency of CD8 + T-cells (Fig. 5c) was observed, accompanied by the upregulation of multiple T

cell-related genes, including CD8A (Fig. 5f). The impact of Afatinib treatment on IFN pathway and MHC family members was also analyzed. The expression of HLA-E/HLA-DRA was upregulated (Fig. 5g), and significant enrichment of IFNG pathway were observed in responsive tumor samples (Supplementary Fig. 2f), which may account for the increased T cell infiltration[22] after neoadjuvant treatment. In addition to CD8 + T-cells, endothelial cells and fibroblasts (Fig. 5c−e) showed increased infiltration after neoadjuvant treatment. Tumoral high-endothelial venules (TU-HEVs)[23] with high SELP and VWF expression were previously identified, and SELP was associated with vascular normalization and immune infiltration[24]. After neoadjuvant treatment, responders exhibited higher SELP and VWF expression in tumor than non-responders (Supplementary Fig. 2g, h), and the main components of TU-HEV endothelial cells may attribute to high immune infiltration. Fibroblasts were a heterogeneous population, and a previous study revealed ADH1B + CAF mainly originated from the CCL19-expressing ADH1B+ cells specifically found in TLS[25] in lung cancer. High ADH1B (Supplementary Fig. 2i) but low FAP (Supplementary Fig. 2j) expression after neoadjuvant treatment might indicate the main contribution of ADH1B + CAF to fibroblasts, potentially enhancing anti-tumor immune activity. Moreover, in the analysis for immune repertoire, an increasing tendency of T-cell receptor (TCR) reads fraction and unique TCR reads was observed after Afatinib treatment (Fig. 5h, i). The evaluation of immune cells and immune repertoire in matched tumor samples showed rough consistency with the above results (Supplementary Fig. 3a−d). Additionally, treatment-associated clones (Supplementary Fig. 3e) were identified in responders, and responsive patients showed relatively high Jaccard and Morisita indices (Supplementary Fig. 3f, g).

In the analysis of LN samples from responders, an increasing tendency of B-cells (Supplementary Fig. 4a) was observed after Afatinib treatment, accompanied by higher expression of MHC-II genes (Supplementary Fig. 4b). Furthermore, B-cell receptor (BCR) reads fraction and unique BCR reads showed an increasing tendency after Afatinib treatment (Supplementary Fig. 4c, d).

**TME differences after Afatinib treatment.** Finally, we focused on the changes in the tumor microenvironment (TME) between different response groups after Afatinib treatment. Several immune cells including T-cells, cytotoxic lymphocytes, B-cells, and NK cells were higher in responders (Supplementary Fig. 5a). Specifically, CD4 + T-cell infiltration was significantly higher in responsive tumors (Supplementary Fig. 5b). In accordance with this, TCR reads fraction, unique TCR reads and BCR clonality exhibited an increasing tendency in responsive tumors (Supplementary Fig. 5c−e).

## Discussion
For advanced NSCLC + , *EGFR*-TKIs have been recommended as first-line treatment because they could significantly improve PFS compared

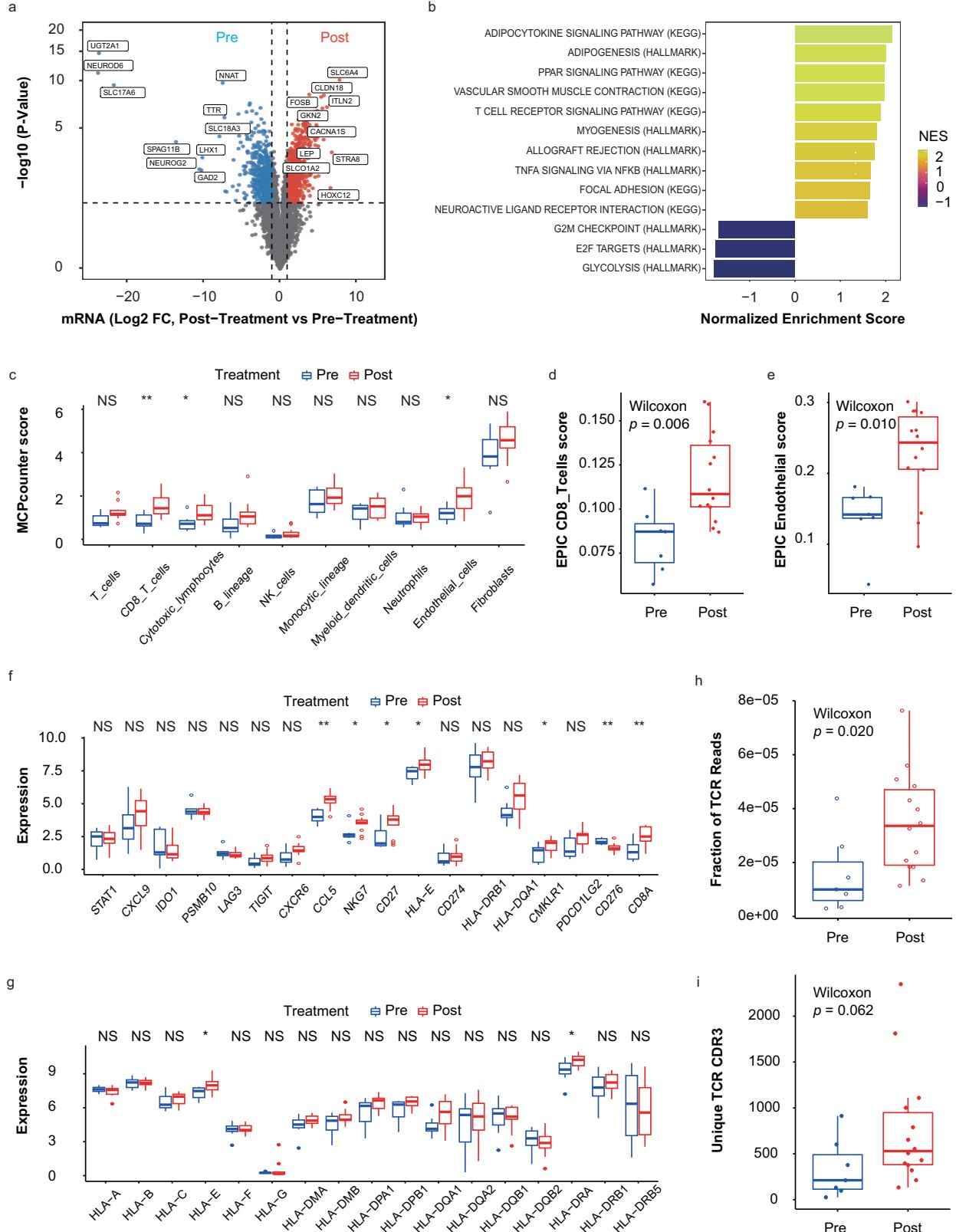

with standard chemotherapy[26–28]. However, the OS benefits of EGFR-TKIs in these patients is limited[29]. Limited studies have focused on the efficacy of neoadjuvant EGFR-TKIs therapy in locally advanced NSCLCm+. Recently published studies have demonstrated that neoadjuvant first-generation EGFR-TKIs in locally advanced NSCLCm+ led to an ORRs of approximately 50%, and significantly improved PFS compared with standard chemotherapy[12–14]. In addition, the

prospective clinical trials (NEOS[30] and NeoADAURA[31]) have shown that neoadjuvant therapy of the third-generation EGFR-TKI (Osimertinib) treatment in locally advanced NSCLCm+ resulted in better ORR (around 70%) than the first-generation TKIs. This phase II prospective trial aimed to fill the gap in assessing the feasibility of the second-generation EGFR-TKIs for locally advanced NSCLCm+. The results demonstrated that NAT achieved a favorable tumor response rate

**Fig. 5 | The changes of tumor microenvironment between baseline and post-treatment tumor samples in responders. a** Differential expression between baseline ($n = 7$) and post-treatment tumor samples ($n = 14$). **b** GSEA analysis between baseline ($n = 7$) and post-treatment tumor samples ($n = 14$). **c** Increasing CD8 + T cells and cytotoxic lymphocytes in post-treatment tumor samples ($n = 14$) compared with baseline ($n = 7$) via MCPcounter, $p = 0.067$, $p = 0.0042$, $p = 0.016$, $p = 0.079$, $p = 0.29$, $p = 0.32$, $p = 0.32$, $p = 0.74$, $p = 0.016$, $p = 0.17$ from left to right. Increasing CD8 + T cells (**d**) and endothelial cells (**e**) in post-treatment tumor samples ($n = 14$) compared with baseline ($n = 7$) via EPIC. **f** Increasing T cell related markers in post-treatment tumor samples ($n = 14$) compared with baseline ($n = 7$), $p = 0.8$, $p = 0.25$, $p = 0.4$, $p = 0.86$, $p = 0.64$, $p = 0.11$, $p = 0.067$, $p = 0.001$, $p = 0.038$, $p = 0.038$, $p = 0.031$, $p = 0.44$, $p = 0.64$, $p = 0.094$, $p = 0.046$, $p = 0.11$, $p = 0.0016$, $p = 0.0056$ from left to right. **g** Increasing MHC members in post-treatment tumor samples ($n = 14$) compared with baseline ($n = 7$). **h** Increasing fractions of TCR reads and (**i**) Unique TCR CDR3 in post-treatment tumor samples ($n = 14$) compared with baseline ($n = 7$), $p = 0.49$, $p = 0.91$, $p = 0.29$, $p = 0.031$, $p = 0.8$, $p = 0.25$, $p = 0.29$, $p = 0.29$, $p = 0.11$, $p = 0.094$, $p = 0.094$, $p = 0.64$, $p = 0.91$, $p = 0.64$, $p = 0.046$, $p = 0.64$, $p = 0.8$ from left to right. Centers, boxes, whiskers, and dots indicate medians, quantiles, minima/maxima, and outliers, respectively. Two-sided Wilcoxon rank sum test was used for comparison in (**c–i**). NS: $p \geq 0.05$; *: $p < 0.05$; **: $p < 0.01$ in (**c**, **f**, and **g**). Source data are provided as a Source Data file. NS: no significance.

(ORR: 70.2%) with an acceptable toxicity profile in patients with locally advanced NSCLCm+. Additionally, considering the mechanism of *EGFR*-TKI resistance is unclear, the results of this trial suggested the opportunity for recurrent patients to continue using the third-generation *EGFR*-TKIs.

To our knowledge, the ORR of locally advanced NSCLCm+ treated with first-generation *EGFR*-TKIs has been observed to be lower than that of advanced NSCLCm+. This could be attributed to two primary reasons. Firstly, the limited duration (6-8 weeks) of neoadjuvant therapy in previous research might be not sufficient for a robust tumor response or complete absorption of necrotic tissue[13]. Secondly, advanced NSCLCm+ may exhibit different tumor-biological behavior compared to NSCLCm+ in a locally advanced stage, potentially rendering tumor cells in the advanced stage more sensitive to *EGFR*-TKIs[32]. However, in this study, locally advanced NSCLCm+ after Afatinib treatment showed a similar ORR as observed in advanced NSCLCm+. This suggests that Afatinib may have earlier anti-oncologic onset and a greater therapeutic effect compared to first-generation *EGFR*-TKIs. These advantages support the suitability of Afatinib for neoadjuvant therapy. Interestingly, the time-dependent Cox model revealed a significant correlation between the EFS and the proportion of SLD reduction after neoadjuvant therapy. Therefore, ORR may serve as a predictor of EFS after neoadjuvant second-generation *EGFR*-TKI treatment.

Furthermore, the number of NAT cycles and the role of surgery in inoperable stage III patients are crucial issues. The optimal number of neoadjuvant treatment cycle need to be addressed for inoperable stage III patients. In this trial, patients who received more than 2 cycles of neoadjuvant therapy demonstrated higher ORR (94% vs 69%), greater radiological regression of targeted lesions (45% vs 39%), and a higher rate of pathological tumor regression (59% vs 41%) compared to those who received only 2 cycles of NAT. These results suggest that extending the duration of NAT may improve efficacy. There is ongoing debate regarding surgical treatment for inoperable NSCLC patients who show downstaging after induction therapy, even if salvage surgery can improve the prognosis of these patients[7,33,34].

Based on the findings of the CheckMate-816 study[35], neoadjuvant treatment has been established as the standard therapy for locally advanced NSCLC. Furthermore, achieving MPR has been identified as a crucial prognostic factor for patients. In the treatment of NSCLCm+ with neoadjuvant EGFR-TKIs, previous studies, and the current study did not observe a significant difference in MPR among patients receiving different *EGFR*-TKIs. This may be attributed to the fact that *EGFR*-TKIs primarily exert cytostatic activity rather than cytotoxicity. As an inhibitor of lung cancer, *EGFR*-TKIs have a limited role in achieving a cure for NSCLCm+. In light of this, improving the prognosis of NSCLCm+, particularly OS, becomes a crucial focus for future research endeavors.

Based on the distinct responses observed in Ex19Del and L858R NSCLC, they can be considered as two different diseases[36,37]. The results from LUX-Lung 3 and LUX-Lung 6 trials have shown that Afatinib provides OS benefits compared to chemotherapy in patients with Ex19Del mutation, but not in patients with L858R mutation[36–38]. In this

trial, 5 participants harboring Ex19Del mutation were excluded from the comparison of Afatinib efficacy between Ex19Del and L858R mutations. The exclusions were due to factors: extremely low mutant abundance for two participants, primary Afatinib resistance for another two participants, and absence of radiological evaluation after NAT for the last participant. No significant differences were observed between Ex19Del and L858R mutations in terms of ORR (78% vs 81%, $p = 0.820$), regression rate of targeted lesions on radiological assessment (45% vs 39%, $p = 0.320$), complete resection rate (83% vs 91%, $p = 0.507$), MPR rate (11% vs 9%, $p = 0.684$), and mean percentage of residual tumor cells on pathological evaluation (54% vs 55%, $p = 0.896$). These results might demonstrate that Afatinib is probably both suitable for both Ex19Del and L858R mutations.

The T790M mutation has been identified as the primary mechanism of secondary resistance to first and second-generation *EGFR*-TKIs[39–42]. The GioTag study demonstrated superior OS in NSCLCm+ who received sequential Afatinib-to-Osimertinib treatment[43,44]. Previous researches has shown that *EGFR*-TKI treatment could affect the expression of PD-L1, influence immune-related pathways, and enhance antigen presentation of B-cells and MHC-II family members[45–47]. In this trial, an increasing tendency in T cell abundance, TCR reads fraction, and unique TCR reads were observed in tumor samples from responders. Additionally, LN samples from responders showed an increase in B-cell infiltration, BCR reads fraction, unique BCR reads, and higher expression of MHC-II family members. The elevated B-cell infiltration, and increased expression of MHC-II members after NAT suggested the activation of antigen presentation pathway in B-cell, which may contribute to the enhanced effectiveness of TKIs in these patients[48]. These findings suggest a positive role of Afatinib in inducing the accumulation of T-cell and B-cell-related features. Therefore, NSCLCm+ patients with secondary resistance to Afatinib may exhibit increased immune features, which could also benefit from immune-checkpoint inhibitors (ICIs).

ICIs-based immune therapy has become the standard therapy for advanced and locally advanced NSCLC, regardless of immune status such as PD-L1 expression and TME characteristics, and has shown excellent efficacy when combined with chemotherapy[49]. However, in the case of NSCLCm+, clinical trials exploring the combination of EGFR-TKIs and ICIs have not demonstrated synergistic effects[50]. These results highlight the challenges associated with the use of ICIs for NSCLCm+. Nevertheless, there are several factors that suggest the potential for ICIs treatment after EGFR-TKI therapy in NSCLCm+. Firstly, EGFR-TKI treatment has been associated with higher tumor mutational burden (TMB)[51,52]. Additionally, there is evidence of elevated PD-L1 expression and changes in the TME after Afatinib treatment, characterized by increased immune cell infiltration and alterations in the immune repertoire. These findings may provide a rationale for considering ICIs treatment following EGFR-TKI therapy for NSCLCm+.

Primary resistance to *EGFR*-TKIs is relatively rare in NSCLCm+. In 2013, Byers and colleagues demonstrated that EMT signature predicted resistance to EGFR-TKI[53]. In this trial, EMT signaling was significantly enriched in non-responders after Afatinib treatment. In

addition, there was increased expression of VEGFB in non-responder after treatment. VEGFB is a member of the vascular endothelial growth factor receptor family[54] and was involved in embryonic angiogenesis[55]. Combinating anti-angiogenic agents with Afatinib may be a potential treatment strategy, particularly for those non-responders[56].

We identified CISH as a molecular biomarker for predicting Afatinib response. Previous studies have reported diverse functions of CISH in different cell types. In T lymphocytes, CISH is induced after TCR stimulation and has been implicated as a good prognostic factor in triple-negative breast cancer[57,58]. In dendritic cell (DC), CISH is predominantly induced during DC development and plays a crucial role in type 1 DC development and DC-mediated T-cell activation[59]. However, CISH has negative negative functions in natural killer (NK) cells. It's reported that CISH is a key negative regulator of interleukin-15 (IL-15) signaling in NK cells, and can induce NK cell exhaustion through regulating metabolic activity[60,61]. Although CISH is well studied in immune cells, its role in tumor cells remains poorly understood. The mechanism by which CISH in tumor cells promotes the sensitivity to Afatinib remains to be explored. We will keep focus on it in future studies.

This study has several limitations that should be acknowledged. Firstly, being a single-arm study, it was not possible to directly compare the efficacy of neoadjuvant treatment between Afatinib and other regimens. A randomized controlled trial with a larger sample size would be necessary for such comparisons. Secondly, being a single-center study, the evidence generated may be limited in its generalizability, and external validation through multicenter research is warranted to confirm the findings. Thirdly, the regimens of adjuvant therapy after complete resection have not been explored in locally advanced NSCLCm+, and this could potentially impact patient prognosis. Fourthly, due to the relatively short follow-up period, the relationship between ORR and prognosis could not be fully assessed. Fifthly, the sample size of paired longitudinal RNA-seq data is small. Therefore, the findings regarding TME changes need to be confirmed by other independent studies. Last, further investigations using WES and TCR sequencing would provide additional insights and should be considered in future studies. Overall, while the results are promising, the efficacy of neoadjuvant Afatinib therapy awaits validation with on long-term follow-up data, which is an important aspect for future research.

In conclusion, Afatinib as a single-agent neoadjuvant therapy for stage III *EGFR*-mutant NSCLC offers a favorable objective tumor response and has an acceptable toxicity profile in clinical practice. Dynamic changes in the tumor microenvironment are observed, particularly in responders, which may have implications for identifying predictive markers for *EGFR*-TKI treatment and guiding the future clinical trials.

## Methods

### Study design
This single-arm, open-label, single-center, phase II clinical trial was conducted at Tongji University affiliated Shanghai Pulmonary Hospital, which was registered at https://www.clinicaltrials.gov before participant enrollment (NCT04201756). The study was conducted in accordance with the Declaration of Helsinki (as revised in 2013). The study was approved by the independent ethic committee in Tongji University affiliated Shanghai Pulmonary Hospital (19229XW). Informed consents were obtained from all participants. The study protocol is provided in the Supplementary Information file.

### Patient eligibility
Treatment naive NSCLCm+ with stage III, older than 18 years, with an Eastern Cooperative Oncology Group (ECOG) performance status of 0 or 1, were enrolled in this study between July 20, 2020, and February 10, 2022. The pathological diagnosis and *EGFR*-mutant status were confirmed by endobronchial ultrasound (EBUS) or percutaneous lung biopsy. The clinical staging was confirmed by positron emission tomography/computed tomography (PET/CT), chest computed tomography (CT), and brain magnetic resonance imaging (MRI). Patients were required to have normal hematological indexes, qualified hepatic, renal, and pulmonary functions so that neoadjuvant therapy followed by radical resection could be tolerated. Based on PET/CT and chest CT at diagnostic time, patients were divided into resectable and potentially resectable subgroups.

Pregnant or breast-feeding patients, patients with unstable systemic disease (interstitial lung disease, pulmonary fibrosis, cardiovascular disease and so on), patients with any anticancer therapy outsides of this trial (*EGFR*-TKIs, chemotherapies, immunotherapies, and so on), and patients with exon 20 codon p.Thr790Met point mutation (T790M) or exon 20 insertion mutation (Ex20Ins) were ineligible for this study. The above examinations and evaluation should be performed within 28 days before the beginning of neoadjuvant therapy.

### Procedures
Stage III NSCLCm+ received 2 to 4 cycles (each cycle lasted 4 weeks) administration of neoadjuvant Afatinib (Giotrif ®, Boehringer-Ingelheim Pharma GmbH, Ingelheim, Germany) therapy (NAT) (oral Afatinib 40 mg, once-daily). Chest CT was performed at the 8th and/or 16th weeks after Afatinib treatment. All CT scans of participants were assessed by the same radiologist and 3 experienced thoracic surgeons via Response Evaluation Criteria in Solid Tumors measurement criteria (RECIST, version 1.1). Radiologically PR or responder were defined as at least a 30% decrease in the sum of diameters of target lesions versus baseline[62]. If PR or SD participants after NAT could be resected completely, surgery should be performed within 3 weeks after Afatinib discontinuation. However, if participants with PR or SD who were still unable to perform radical resection after 16 weeks NAT, or participants with PD during neoadjuvant phase, they should consider withdrawing from this study and treating them based on multi-disciplinary therapy (MDT).

### Efficacy and safety assessment
The primary endpoint was ORR accessed by the RECIST criteria. The secondary endpoints were MPR rate (proportion of patients with no more than 10% residual viable tumor cells), pCR rate (proportion of patients without residual viable tumor cell)[63], the complete resection rate, EFS, PSF, DFS, OS, and TRAEs. By the data cutoff date on May 1st, 2023, the survival data were immature. In this report, EFS was calculated and reported, because compared with DFS and PFS, EFS can better reflect the efficacy of systematic treatment.

In addition, some participants refused or failed to fill in the health-related life quality questionnaires as required due to personal reasons or privacy issues, which affected the data quality of this part. In this case, the analysis of the predefined health-related quality of life questionnaires was abandoned in this report.

### Sample size estimation
The primary endpoint of this research was ORR, which was taken to calculate the sample size. Previous studies reported that the ORR of neoadjuvant Erlotinib for NSCLC (EMERGING-CTONG 1103) at stage III-N2 was 54.1%[12]. The ORR in the control group for neoadjuvant chemotherapy was 34.3%, and the ORR of neoadjuvant Afatinib for NSCLC (ASCENT) at stage III was 75%. Given that the small sample size of ASCENT trial (*n* = 22) may lead to overestimation of the ORR rate, and that Afatinib had a better therapeutic effect as the generation-II TKI than the generation-I TKIs, so the ORR rate was estimated to be 60% in this study. The ORR rate was expected to be 60% in this study, and the neoadjuvant chemotherapy group of EMERGING trial was taken as the historical control group. It was calculated with $\alpha = 0.05$ (two-tailed) and the power of test $(1-\beta) = 90\%$. According to the sample calculation of the one-sample rate test, when $P_0 = 34\%$, and $P_1 = 60\%$, the sample

size will be 42 as calculated. With a drop-off rate of 10%, the total sample size will be 47.

## Follow-up

Follow-up was achieved by telephone contact or outpatient visit. For surgical treated participants, chest CT were performed every 3 months for the first year postoperatively, every 6 months for 2 to 5 years, and annually from then on. MRI of the brain, ultrasonography of abdominal regions, bone scans were performed annually. The follow-up was conducted until death. Local recurrence was defined as the recurrence in the primary site or mediastinal lymph nodes, while distant recurrence was defined as recurrence in other sites.

## RNA sequencing and data processing

Total RNA from fresh frozen tissues was extracted with TRIzol. Sequencing libraries were generated using a NEBNext Ultra RNA Library Prep Kit for Illumina, and index codes were added to attribute sequences to each sample. The libraries were pooled, and paired-end sequencing (2×150 bp reads) was performed using an Illumina NovaSeq 6000. After RNA sequencing, raw fastq files were trimmed via fastp (v0.20.1)[64] and aligned to GRCh38 reference genome by STAR (v2.7.6a)[65] with default settings. After obtaining the BAM files, read counts were summarized by featureCounts (v2.0.1)[66] and TPMs (Trans Per Million) were generated using Salmon (v0.6.0)[67]. Batch effects were adjusted using "combat" function in sva package.

## Differentially expressed gene analysis

We used DESeq2 to calculate differential gene expression between sample groups[68]. The DESeq2 profiles genes according to model gene count expression data and calculates log2 fold change, which estimates the effect size and represents gene changes between comparison groups. The two-sided Wald-test statistics are computed to examine the differential expression across the comparison groups. Genes with |Log2Fold Change | > 1 and Wald test $p < 0.05$ were defined as differentially expressed genes. We used the volcano plots to visualize the differential gene expression results.

## Gene set enrichment analysis (GSEA)

In GSEA analysis, results for all protein-coding genes were ranked by log2 fold change and evaluated with the 'GSEA' algorithm[69]. 'Hallmark' and 'KEGG' gene sets were acquired from MSigDb. We filtered GSEA results based on the criteria that the $p$-value < 0.05 and visualized the pathways as candidates based on the normalized enrichment score.

## Tumor micro-environment estimation

The immune scores of each sample were calculated using the "ESTI-MATE" R package[70]. The infiltration of multiple immune cells was evaluated by "mcpcounter (v1.2.0)"[71] and "EPIC (v1.1.5)"[72]. For immune repertoire analysis, TRUST4 (v1.0.0) algorithm was applied to evaluate the immune repertoire and to extract T and B cell receptor (TCR and BCR, respectively) complementarity-determining region 3 (CDR3) sequences[73,74].

## Immunohistochemistry analysis

Tumor samples were fixed with formalin and embedded in paraffin. The tumor samples were provided by the Pathology Department of Shanghai Pulmonary Hospital. Stain the slide with Anti-CISH/CIS antibody-C-terminal (Abcam: ab191447, dilusion:1/100) and Anti-PD-L1 antibody (PD-L1: E1L3, Rabbit mAb, CST: 13684 S, dilusion:1/200). The immunohistochemical (IHC) score was calculated according to the proportion of positive tumor cells (the proportion less than 1% was 1; the proportion from 1%-49% was 2; the proportion larger than 50% was 3) and the average intensity of positive staining (negative staining was 1; weak staining was 2; and strong staining was 3). The range of IHC score was from 1 to 9 for each sample.

## Statistical analysis

Efficacy and safety analyses were based on the intention-to-treat (ITT) population who received NAT at least one dose. Descriptive statistics, chi-squares test, Fisher's exact test, independent sample $t$-test, Kaplan-Meier method, Log-Rank test, and Cox proportional hazards regression model were used. The median follow-up time was calculated by the reverse Kaplan-Meier method. EFS and OS were calculated and reported in this work to evaluate the feasibility of neoadjuvant treatment. EFS was defined as the time between the first dosage of neoadjuvant Afatinib and any progression of the disease before or after surgery, disease recurrence after surgery, disease progression in the absence of surgery, or death from any cause. OS was defined as the time from the date of diagnosis to death from any cause. The relationship between EFS and the parameters of participants that should be evaluated after neoadjuvant treatment was assessed by the time-dependent Cox model. For genomic biomarker analysis, Wilcoxon rank sum test was applied to compare the difference between sample groups independently. To separate patients into low- or high-*CISH* groups, the cutoff was generated based on the association between *CISH* expression and survival data using the survminer R package. All $p$ values were two-sided, with $p < 0.05$ considered statistically significant. Clinical data were analyzed by SPSS software (version 26.0, IBM Corp, Armonk, NY), and exploratory analyses were conducted using R programing (version 4.1.0).

## Reporting summary

Further information on research design is available in the Nature Portfolio Reporting Summary linked to this article.

## Data availability

The raw RNA-seq data generated in this study have been deposited in the Genome Sequence Archive of the BIG Data Center at the Beijing Institute of Genomics, Chinese Academy of Science, under accession code HRA003549. The sequencing data are available under controlled access due to data privacy laws related to patient consent for data sharing and the data should be used for research purposes only. Access can be obtained by approval via the Data Access Committee in the GSA-human database. The approximate response time for accession requests is about 2 weeks. Once access has been granted, the data will be available to download for 3 months. Clinical data are not publicly available due to involving patient privacy, but can be accessed from the corresponding author Peng Zhang (Email: zhangpeng1121@tong-ji.edu.cn), upon request for 3 years; individual de-identified patient data will be shared for clinical study analyses. The study protocol is available in the Supplementary Information file. Data from two publicly available datasets were incorporated into our study: TCGA-LUAD level three RNA-seq data and clinical information from patients were acquired from the UCSC Xena website (https://xenabrowser.net/). Cell lines annotated as "Non-Small Cell Lung Cancer (NSCLC), Adeno-carcinoma" with *EGFR* mutations from DepMap (https://depmap.org/) were applied with molecular profiles and drug response information used. Pearson correlation was conducted between CISH expression and drug responses information to Afatinib in PRISM secondary screen (https://depmap.org/portal/prism/) and CTRP database (https://portals.broadinstitute.org/ctrp). Source data are provided with this paper. The remaining data are available within the Article, Supplementary Information or Source Data file. Source data are provided with this paper.

## Code availability

The code used in this study has been deposited at https://github.com/zero123321/Neoadjuvant-Afatinib.

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

## Acknowledgements

We thank all participants and their relatives for being a part of this trial. This work was supported by Boehringer-Ingelheim Pharma GmbH, Ingelheim, Germany, the National Natural Science Foundation of China (Grant No. 82125001), Innovation Program of Shanghai Municipal Education Commission (Grant No. 2023ZKZD33), and Clinical Research foundation of ShangHai Pulmonary Hospital (grant No. FKLY20004). The sponsor and funders had no role in study design, data collection, data analysis, or manuscript writing.

## Author contributions

P.Z., and D.B. conceived and designed the study. W.H., L.D., D.Z., and N.S. provided the materials and patients support. G.J. and Y.Z. provided the administrative support. D.B., H.X., X.Z., F.S., Y.X., Z.H., Y.H., D.Z., J.Y., X.B., W.W., and J.Huang. collected the data. D.B., L.S., J.Hu., and L.Z. analyzed and interpreted the data. D.B. and L.S. wrote the first draft of the manuscript. All authors approved the final version of the manuscript for submission.

## Competing interests

The authors declare no competing interests.

## Additional information

[1]Department of Thoracic Surgery, Shanghai Pulmonary Hospital, Tongji University School of Medicine, Shanghai 200433, China. [2]Central Laboratory, Shanghai Pulmonary Hospital, School of Medicine, Tongji University, Shanghai 200433, China. [3]Department of Animal Experimental Center, Shanghai Pulmonary Hospital, Tongji University School of Medicine, Shanghai 200433, China. [4]Nanchang University School of Medicine, Nanchang, Jiangxi 330006, China. [5]Department of Radiology, Shanghai Pulmonary Hospital, Tongji University School of Medicine, Shanghai 200433, China. [6]Department of Molecular Pathology, Shanghai Pulmonary Hospital, Tongji University School of Medicine, Shanghai 200433, China. [7]Department of Clinical Research Center, Shanghai Pulmonary Hospital, Tongji University School of Medicine, Shanghai 200433, China. [8]Wenzhou Medical University, Wenzhou, Zhejiang 325035, China. [9]Shihezi University School of Medicine, Shihezi, Xinjiang 832099, China. [10]These authors contributed equally: Dongliang Bian, Liangdong Sun, Junjie Hu, Liang Duan, Haoran Xia. [11]These authors jointly supervised this work: Wenxin He, Yuming Zhu, Gening Jiang, Peng Zhang. ✉e-mail: 0wenxinhe@tongji.edu.cn; ymzhu2005@aliyun.com; geningjiang@tongji.edu.cn; zhangpeng1121@tongji.edu.cn

