## [Peer Review File · Nature Communications]

Neoadjuvant Afatinib for Stage III EGFR-Mutant Non-Small Cell Lung Cancer: A Phase II StudyREVIEWER COMMENTS

Reviewer #1 (Remarks to the Author): with expertise in biostatistics, clinical trial study design

This is a phase II study to evaluate Afatinib in NSCLC with n=47. The efficacy summary results looks good with ORR=70% which is higher than expected 60%. The statistical approaches used for summary statistics looks good in general, while some details need to be specified, such as (1) which methods were used to calculate median time to follow up (usually reverse Kaplan-Meier), (2) which Wilcoxon tests were used for which type of analysis, e.g., for paired data, it's Wilcoxon signed rank test, for two independent group comparison, it's Wilcoxon rank sum test, (3) need 95% confidence interval for key results, such as ORR. (4) need to specify the ORR rate for the first 20 subjects with 95% CI. It's a requirement based on the stat section in the protocol.

While I have some major stat questions:

a) how to define EFS ? In the attached protocol, the EFS definition is not clear how it is different from PFS. Please confirm how a subject might be censored if alive and not progressed. Is receiving surgery an event? Need to clarify

b) There are major statistical issues for figure 3(e)(f). I assume that the "surgical treatment" performed after the study entry, similar as the ORR will be evaluated after the study entry. That's future information, so it's not appropriate to use this future information to classify patients' status at day 0 (first dose of Afatinib). Kaplan Meier curve is not appropriate to show two group comparisons if the group defined by the variable happened after study entry (we called it Naive KM). For example, Figure 3e showed patients with surgery have better EFS than those without surgery, which actually might not be true. All patients entering the study are actually in the "non-surgery" group, and some of them received surgery later, then they will jump to another curve called "surgery". The appropriate approach is to use the time-dependent Cox model instead (and delete KM curves).

Similarly, Figure 3f is defined using future response information (ORR), thus it has a similar issue. While it might be OK to use a landmark analysis to generate the KM curve and day 0

will be re-defined as the day of ORR evaluation.

All in all, it would be good to invite a biostatistician to review the stat analysis section and re-build the time-dependent Cox model for 3(e) (Need to remove 3(e)) and 3(f). For 3(f), it might be OK to use landmark analysis. Or just delete 3(e) and 3(f). Only report the results from the time-dependent Cox model, respectively.

Please find some references as below

- Therneau, T., Crowson, C., & Atkinson, E. (2013). Using time dependent covariates and time dependent coefficients in the cox model. *Red*, 2(1).
- Schultz, L. R., Peterson, E. L., & Breslau, N. (2002). Graphing survival curve estimates for time-dependent covariates. *International journal of methods in psychiatric research*, 11(2), 68-74.
- Bernasconi, D. P., Valsecchi, M. G., & Antolini, L. (2018). Non-parametric estimation of survival probabilities with a time-dependent exposure switch: application to (simulated) heart transplant data. *Epidemiology, Biostatistics and Public Health*, 15(3).

Reviewer #2 (Remarks to the Author): with expertise in lung cancer, cancer immunology

In this manuscript, Bian and colleagues describe a Phase II study of Afatinib in the neoadjuvant setting for the treatment of stage III EGFR-mutant NSCLC. The authors show an overall response rate of 70.2% and a major pathological response rate of 9.1%. The authors also perform RNA sequencing on longitudinal tissue in a subset of patients to assess changes in tumor microenvironment, immune infiltration and T cell/B cell repertoire. Overall, the study is of interest. However, the following issues should be addressed.

Major issues

1-Analysis of the T and B cell repertoire is a strength but should be expanded upon. If available, TCR sequencing should be performed to ensure capture of the entire repertoire, as should whole exome sequencing to identify changes in TMB/neoantigens/allele fraction.

2-Existing analysis of the T and B cell repertoire should be expanded in patients with matched longitudinal samples. This includes analysis of clonality where feasible, as well as changes in the T cell repertoire (Jaccard Index/Morisita Index) and clone tracking, among others, between tumors and LNs.

3-Tissue staining should be performed where available to determine whether CISH is expressed in tumor or immune cells.

Minor issues

4-Longitudinal data should be presented as such where available with connecting lines to identify treatment-associated changes rather than solely overall differences between groups. Likewise, trends pre/post should be compared between responders and non-responders.

5-Differences in endothelial cells and fibroblasts observed by MCPcounter should be elaborated upon/discussed.

6-Inclusion of both relative and absolute immune cell quantifications would be informative.

7-Line 113: a $p=0.217$ is generally considered a “better EFS tendency”, this language should be toned down.

8-Considering the impact of Afatinib on T cells – what are the effects on IFN-g pathway, MHC I and MHC II expression and associated chaperones?

9-In Extended Fig. 2F, it is unclear how you can say T cell activity rather than solely infiltration is higher in responders.

10-A number of axes are either unlabelled (such as Fig. 4) or have unclear axis labels such as “Activity”. Please include units and clarify where applicable.

11-Why are changes in MHC II but not MHC I observed according to Extended Fig. 2B?

12-Please include PD-L1 status where available and assess impact on clinical and pathological responses.

13-p-values are missing for certain plots, such as Fig. 4D/E/H/I.

14-What role did adjuvant therapies (chemo-IO vs targeted) play in clinical outcomes? This should be elaborated upon.

15-It is unclear what the difference is between the data presented in Fig. 5C and Extended Fig. 2A.

16-The proposed mechanism accounting for increased T cell infiltration with neoadjuvant Afatinib should be elaborated upon. What about expression of IFN pathway genes, CXCL9/10/11, MHC chaperones, etc?

Reviewer #3 (Remarks to the Author): with expertise in lung cancer

There are multiple treatment strategies in L/A NSCLC with activating EGFR mutation depending on the tumor size, LN, or operability as follows: 1> upfront surgery ◇ postoperative adjuvant platinum-based chemotherapy followed by adjuvant osimertinib, 2> neoadjuvant chemotherapy followed by surgery, 3>definitive concurrent chemo radiation therapy, 4> upfront EGFR-TKI.

The author conducted neoadjuvant Afatinib phase II trial with 47 stage III NSCLC to investigate the its efficacy in neoadjuvant setting.

This samples size is too small to identify the efficacy of neoadjuvant afatinib in stage III NSCLC, which were composed of multiple stage (including IIIA-IIIC). The treatment strategies of stage III group would not be unified and their prognosis is so heterogeneous.

Therefore, I think it is unreasonable to evaluate the effect of neoadjuvant afatinib with 47 people in stage III, where these various diseases are consisted.

Is there any prespecified post op adjuvant chemotherapy in study protocol? I think that various post op adjuvant treatment in this study could be compounding factors to interpret the survival outcome. Especially, this study enrolled EGFR m+ NSCLC patients. Therefore, adjuvant immunotherapy would be not generally used standard adjuvant therapy in these patients. We found two patients treated with adjuvant therapy in these patients.

Of potentially resectable patients (n=25), approximately half of patients (n=11) did not receive the surgery. The fact that potentially resectable patients did not undergo surgery in half of the patients would be interpreted as not satisfying the goal of neoadjuvant therapy. Or, I think that the initial evaluation of resectability in these patients was not appropriately evaluated.

The patients achieved mPR or pCR based on the pathology response after neoadjuvant shows excellent survival outcomes. This is generally accepted concept. I suggest the additional analysis in these patients according to the pathology response.

Reviewer #4 (Remarks to the Author): with expertise in lung cancer

Dear Dr Bian and colleagues, it was a great pleasure to review this interesting study which propose the utility of neoadjuvant afatinib in stage III EGFR mutant non-small cell lung cancer.

Major comments

1) Historically we know that TKIs had a predominant cytostatic activity rather than cytotoxic, and previous works have suggested this kind of therapy just delayed disease recurrence but not cured patients, which is one of the main objectives in this context. I consider adding something about this to your discussion. ¿Would you consider adding chemotherapy in combination with TKIs in this context? Why don't you comment anything about the current phase 3 trial with osimertinib, which is evaluating this issue? What about the increased risk of micrometastatic disease at SNC in stage III patients and the increased risk of recurrence at this site?

2) Please I suggest adding something in the discussion about the low rate of major pathologic responses and complete pathologic responses which have been experienced in all EGFR neoadjuvant studies including yours. This could be put in context with this strategy, considering the benefits of other kinds of treatments like immune checkpoint inhibitors. This was in line with my previous comment. Do we really cure more patients with TKIs?

3) Currently the upfront line treatment in the EGFR metastatic setting is osimertinib due to an increased CNS activity and a more prolonged PFS compared with first-generation TKIs in this population. Furthermore, the main site of recurrence especially in the stage III population is CNS, please justify your decision to opt for a second-generation TKI than a third-generation in your study design. Additionally, why don't you analyze sites of recurrence? It would be interesting to know the proportion of patients who had a CNS recurrence.

4) Bearing in mind that adjuvant therapy is the standard treatment in resectable stage III patients who underwent surgical resection. Why don't you standardize adjuvant therapy in your study? Some patients received EGFR TKI, others chemotherapy, but up to 25% didn't receive any post-surgical therapy. How reliable is the EFS endpoint, with so much variability in the adjuvant setting? Please explain, why in this stage III population up to one-fourth of patients did not receive any adjuvant treatment.

5) Another key point to be discussed is to clarify, why did you choose the objective response rate as the primary endpoint? As it was stated in the introduction, and it was clear in the CTONG 1103 a non-negligible number of patients do not reach tumor response due to the appearance of necrosis which can sub-estimate the downstaging of tumors. Thus, many neoadjuvant studies choose for more solid outcomes like major pathologic response, which in some contexts has been demonstrated to be a surrogate of survival.

6) I like so much the analysis of tumor microenvironment and molecular gene analysis. In this regard, I have some recommendations.

- It will be interesting if it is possible to add an analysis of PD-L1 expression before and after

NAT.

- Second, it was quite interesting the increase in VEGF-B expression after NAT. I suggest discussing these results deeply. Based on these results, would you consider adding antiangiogenics in this context? This analysis opens the window to assess this combination strategy (EGFR TKI with antiangiogenic) to reach better responses.

7) In the differential expression analysis, do you assess TP53?

8) In the ORR table (table 2), patients with less than 60% of residual disease were greater than 50%. In other studies, this cut-off point of the residual disease has suggested better outcomes. Why does this information not discuss? Please explain, why do you choose a cutoff point of 60%? and what is the relevance of this finding in your study? In this regard, don't you consider it interesting to assess an event-free survival curve stratified by this 60% cut-off point?

9) Do you assess the percentage of necrosis in post-treatment samples? And if it is the case, do you explore the correlation between necrosis and response?

Minor comments

1) Between lines 212-216 is suggested that afatinib had a greater efficacy in terms of response rate than first-generation TKIs. I'm not convinced that this can be supported by the cited evidence and their results. Erlotinib in the CTONG 1103 was administered orally for a median of 42 days with an ORR of 53%. In contrast in your study, afatinib was administered until 4 cycles of treatment with a median of 2.7 and an ORR of 70.2%. Moreover, your exploratory analysis clearly shows that patients who received more than two cycles reach better responses (94% vs 69%), thus, differences between first-EGFR TKIs studies and yours could be explained by the number of cycles administered in the previous surgery.

2) Please review lines 143-145 on page 10, it's quite hard to understand. I suggest rewriting this sentence.

3) I suggest replacing the phrase "increasing tendency" in most of the paragraphs, my impression is that this phrase is overused in your manuscript. In addition, I understand that statistical differences did not exist in many comparisons, however, I did not consider the most appropriate way to describe a non-significant difference.

4) BCR is missing in the abbreviation and acronyms list.

5) I suggest in general, improving English. In some fragments of the manuscript is quite difficult to follow the idea.

Response letter of *Nature Communications*

Reviewer #1: with expertise in biostatistics, clinical trial study design

This is a phase II study to evaluate Afatinib in NSCLC with n=47. The efficacy summary results looks good with ORR=70% which is higher than expected 60%. The statistical approaches used for summary statistics looks good in general, while some details need to be specified, such as (1) which methods were used to calculate median time to follow up (usually reverse Kaplan-Meier), (2) which Wilcoxon tests were used for which type of analysis, e.g., for paired data, it's Wilcoxon signed rank test, for two independent group comparison, it's Wilcoxon rank sum test, (3) need 95% confidence interval for key results, such as ORR. (4) need to specify the ORR rate for the first 20 subjects with 95% CI. It's a requirement based on the stat section in the protocol.

Reply: Thank you for your kind suggestions. We indeed should have supplemented and improved the statistical methods according to your comments to make the description of this manuscript seems more accurate and concise. Thank you for your assistants.

- (1) In this trial, we used reverse Kaplan-Meier method to calculate the median follow-up time. And this information has been supplemented to the Methods (Statistical Analysis) in the revised version of the manuscript in the red font:

Line 450-451, Page 24: The median follow-up time was calculated by reverse Kaplan-Meier method.

- (2) In genomic biomarker analysis of this trial, 2 independent sample groups were compared by Wilcoxon rank sum test. And this information has been supplemented to the Methods (Statistical Analysis) in the revised version of the manuscript in the red font:

Line 456-457, Page 24: For genomic biomarker analysis, Wilcoxon rank sum test was applied to compare the difference between sample groups independently.

- (3) We added 95% confidence interval (95% CI) to the presentation of data for the primary and secondary endpoints. And this information is supplemented to the Abstract, and Results (Efficacy) in the revised version of the manuscript in the red font:

Line 40-42, Page 3: The objective response rate was 70.2% (95% confidence interval (CI): 56.5% to 84.0%) which met the pre-specified primary endpoint. The major pathologic response (MPR) rate was 9.1% (95% CI: 1.9% to 24.3%).

Line 103-108, Page 8: The primary endpoint, ORR, was 70.2% (95% confidence interval (CI): 56.5% to 84.0%). For secondary endpoints, the rate of R0 was 87.9% (95% CI: 71.8-96.6%) among the 33 participants who received NAT followed by surgery. Major pathologic response (MPR) was evaluated in these 33 samples. Among the three MPR participants (MPR rate: 9.1%, 95% CI: 1.9% to 24.3%), only one participant achieved pathologic complete response (pCR).

- (4) According to the protocol, of the 20 subjects who were first enrolled, 19 subjects underwent radiological evaluation after neoadjuvant treatment, and 14 of them achieved radiological partial response (14/19). The ORR for the first 20 subjects were 73.7% (95% CI: 48.8%-90.9%). This data did not present in the part of Results as this belongs to the data in the course of trial. We will further refine it in the revised manuscript if you suggest that we need to add it to the manuscript.

While I have some major stat questions:

a) how to define EFS ? In the attached protocol, the EFS definition is not clear how it is different from PFS. Please confirm how a subject might be censored if alive and not progressed. Is receiving surgery an event? Need to clarify

Reply a) Thank you for your kind comments. In this trial, as the same as clinical trial CheckMate 816 ¹, **event-free survival (EFS) was defined as the time from the first dosage Afatinib treatment to any progression of disease precluding surgery, progression or recurrence of disease after surgery, progression of disease in the absence of surgery, or death from any cause.**

Compared with progression-free survival (PFS), event-free survival (EFS) was used as a secondary endpoint could assess the survival status of a subset of patients whose disease progressed during the neoadjuvant treatment phase but still underwent radical surgery.

Besides, according to your comment, if a censored subject alive and not progressed would be considered as event-free. Fortunately, in this prospective clinical trial, from now on, no censoring events occurred.

Last, according to the protocol of this trial, receiving surgery after neoadjuvant treatment should not be defined as an event occurrence.

b) There are major statistical issues for figure 3(e)(f). I assume that the "surgical treatment" performed after the study entry, similar as the ORR will be evaluated after the study entry.

That's future information, so it's not appropriate to use this future information to classify patients' status at day 0 (first dose of Afatinib). Kaplan Meier curve is not appropriate to show two group comparisons if the group defined by the variable happened after study entry (we called it Naive KM). For example, Figure 3e showed patients with surgery have better EFS than those without surgery, which actually might not be true. All patients entering the study are actually in the "non-surgery" group, and some of them received surgery later, then they will jump to another curve called "surgery". The appropriate approach is to use the time-dependent Cox model instead (and delete KM curves).

Similarly, Figure 3f is defined using future response information (ORR), thus it has a similar issue. While it might be OK to use a landmark analysis to generate the KM curve and day 0 will be re-defined as the day of ORR evaluation.

All in all, it would be good to invite a biostatistician to review the stat analysis section and re-build the time-dependent Cox model for 3(e) (Need to remove 3(e)) and 3(f). For 3(f), it might be OK to use landmark analysis. Or just delete 3(e) and 3(f). Only report the results from the time-dependent Cox model, respectively.

Reply b) Thanks for your kind comments, the statistical methods previously applied in this study (Kaplan-Meier Method and Log-Rank Test) which have many disadvantages and bias when the data are not yet mature, and these statistical methods for relationship between ORR/MPR and prognosis will be evaluated in the post hoc analysis.

According to your suggestion, **we use the time-dependent Cox model to evaluate the relationship between EFS and the factors which were evaluated after the study entry (including ORR and treatment procedure). In this trial, we replaced the parameter of ORR by the proportion of reduction in sum of lesion diameter (SLD).** After that, the correlation between reduction of SLD and EFS was established according to neoadjuvant treatment cycle by the time-dependent Cox model. As the Table showed that, reduction of SLD could significantly affect EFS of participants (HR: 0.954, 95% CI: 0.924-0.985, $P=0.004$).

Supplementary Table 3

Variables	HR	95% CI	P
Reduction of SLD (%)	0.954	0.924-0.985	0.004
Pathological response (%)	0.974	0.937-1.014	0.200

SLD: sum of lesion diameter (the longest diameter of the primary lesion and that of the shortest diameter of the lymph node metastasis)

The statistical method you suggest has been updated in the part of Methods (Statistical Analysis) in the revised version of the manuscript in the red font:

Line 454-456, Page 24: The relationship between EFS and parameters of participants that should be evaluated after neoadjuvant treatment was assessed by the time-dependent Cox model.

Besides, the relationship between the reduction of lesion diameter and EFS was recalculated by the time-dependent Cox model, which the results have been supplemented in the part of Results (Survival Analysis) in the revised version of the manuscript in the red font:

Line 122-126, Page 9: According to the evaluation of the time-dependent Cox model, the proportion of radiological reduction in sum of lesion diameter (SLD) could significantly improve EFS of participants received neoadjuvant Afatinib treatment (HR: 0.954, 95% CI: 0.924-0.985, $p = 0.004$). However, the tendency of EFS could not be observed due to pathological response and surgical treatment (Supplementary Table 3).

Finally, we express sincerely thanks to your nice work again. With your help, we avoided oversights regarding statistical issues, making the contents of this study more informative and the conclusions of this study more reliable.

Please find some references as below

- Therneau, T., Crowson, C., & Atkinson, E. (2013). Using time dependent covariates and time dependent coefficients in the cox model. *Red*, 2(1).
- Schultz, L. R., Peterson, E. L., & Breslau, N. (2002). Graphing survival curve estimates for time-dependent covariates. *International journal of methods in psychiatric research*, 11(2), 68-74.
- Bernasconi, D. P., Valsecchi, M. G., & Antolini, L. (2018). Non-parametric estimation of

survival probabilities with a time-dependent exposure switch: application to (simulated) heart transplant data. *Epidemiology, Biostatistics and Public Health*, 15(3).

Reviewer #2: with expertise in lung cancer, cancer immunology

In this manuscript, Bian and colleagues describe a Phase II study of Afatinib in the neoadjuvant setting for the treatment of stage III EGFR-mutant NSCLC. The authors show an overall response rate of 70.2% and a major pathological response rate of 9.1%. The authors also perform RNA sequencing on longitudinal tissue in a subset of patients to assess changes in tumor microenvironment, immune infiltration and T cell/B cell repertoire. Overall, the study is of interest. However, the following issues should be addressed.

Reply: Thank you for your review of this study and kind suggestions. Your suggestions will be of great help to refine our study. What is more, we have also revised the concerning contents in the manuscript according to your suggestions, hoping to make this study more concise and rigorous.

Major issues

1-Analysis of the T and B cell repertoire is a strength but should be expanded upon. If available, TCR sequencing should be performed to ensure capture of the entire repertoire, as should whole exome sequencing to identify changes in TMB/neoantigens/allele fraction.

Reply 1): Thank you for your comment and the additional TCR sequencing and WES maybe helpful to further understand the influence of neoadjuvant treatment to immune repertoire and genomic changes.

As a prospective clinical trial, guaranteeing the safety of participants during diagnosis and treatment is the most important issue. Therefore, at the time of diagnosis, a limited amount of tumor tissue in baseline was obtained. After excluding the tumor samples that need to be used as pathological diagnosis and genomic testing, the remaining baseline tumor tissue for exploratory researches was minimal, which not be sufficient for other sequencing including WES, TCR sequencing after RNA-seq. The relevant limitations were mentioned in the part of limitation in revision manuscript in the red font:

Line 340-343, Page 19: Second, in this single-center study, some evidence was insufficient because of the relatively small sample size especially for longitudinal data in RNA-seq analysis.

Line 346, Page 19: Fifth, further WES and TCR sequencing were needed in future study.

Although NSCLC harboring EGFR mutations has been shown to have a lower TMB compared with tumors without oncogenic drivers ², TMB was found to be significantly increased after EGFR-TKI treatment in some studies ^{2,3}. As TMB and neoantigen load have been shown to be strongly correlated ^{4,5}, we speculated neoantigen load may also be higher after treatment. TMB ⁶ was associated with better immunotherapeutic response and a high TMB could increase neoantigens inducing immune response ⁷, which may indicate the application of ICB after TKI resistance. The relevant content was mentioned in the part of discussion in revision manuscript in the red font:

Line 317-320, Page 17-18: Along with higher TMB after EGFR-TKI, elevated PD-L1 expression, and TME changes after Afatinib treatment with higher immune cell infiltration and immune repertoire may also provide evidence for ICIs treatment after EGFR-TKI treatment.

Fortunately, as you mentioned, the results of TCR and WES are also of interest to us, and we have been carried out clinical trials continually to complete the exploratory contents.

2-Existing analysis of the T and B cell repertoire should be expanded in patients with matched longitudinal samples. This includes analysis of clonality where feasible, as well as changes in the T cell repertoire (Jaccard Index/Morisita Index) and clone tracking, among others, between tumors and LNs.

Reply 2): Thank you for your comment, this is a very meaningful suggestion indeed.

As we all known, the most realistic tendencies of tumor microenvironment remodeling were reflected by contrasting the tumor tissue before and after neoadjuvant therapy in the same participant ⁸. In fact, we have performed immune repertoire analysis in patients with matched longitudinal samples. However, due to the limited sample size in this phase II clinical trial, longitudinal data of responsive and non-responsive tumors, only three and five pairs, respectively, which were two and one for responsive and non-responsive LNs. The relevant tendency of tumor microenvironment was consistent with our previous results, but the statistical significance was not significant. Therefore, we did not include the results for this subset of matched longitudinal samples (as the Reply Fig. 1 shown below) to the manuscript, however, we will further refine it in the revised manuscript if you suggest that we need to add it to the

manuscript. The relevant limitation was mentioned in the part of limitation in revision manuscript in the red font:

Line 340-343, Page 19: Second, in this single-center study, some evidence was insufficient because of the relatively small sample size especially for longitudinal data in RNA-seq analysis. Therefore, external validation by multicenter research should be conducted in the future.

Thank you for your suggestions again, as you mentioned, we have carried out more relevant clinical trials to enroll more participants harboring EGFR mutation to analyze the most realistic tendencies of tumor microenvironment remodeling induced by target-therapy through the matched samples of EGFR mutant patients. These trends of change probably guide the novel treatment in the future. We will be happy to edit the manuscript further based on your meaningful comments.

Reply Fig. 1. Immune repertoire analysis in responder and non-responder patients with matched longitudinal tumor samples.

3-Tissue staining should be performed where available to determine whether CISH is expressed in tumor or immune cells.

Reply 3): Thank you for your comment. We gratefully acknowledge your initial recognition of CISH, which used as a predictor for triple-negative breast cancer, was identified in this trial for EGFR-mutant NSCLC. What is more, the predictive efficacy of CISH in EGFR-mutant NSCLC was validated from the participant in this trial and the cell-lines of preclinical drug experiments.

We have conducted IHC staining for CISH. CISH IHC-score was moderately high in baseline responder tumor samples (Fig. 4f), and baseline CISH IHC expression could predict TKI efficiency (AUC=0.792) (Fig. 4g). In addition, patients with higher CISH expression showed better OS ($p = 0.13$) and EFS ($p = 0.11$) tendency (Fig. 4h-i). Due to limited samples size, the statistical significance was not reached but the IHC analysis results showed consistent tendency with RNA-seq analysis. Taken together, CISH was a promising biomarker for predicting TKI efficiency.

The results were supplemented into the revision manuscript in the red font:

Line 167-177, Page 11: For baseline tumor samples, we found that Cytokine-Inducible SH2-Containing Protein (CISH) was highly expressed in responsive tumor samples at baseline based on RNA-seq (responsive sample vs non-responsive sample: 10 vs 7, $p = 0.007$) (Fig. 4b), and the same tendency was verified in CISH-IHC score (responsive sample vs non-responsive sample: 6 vs 6, $p = 0.100$) (Fig. 4f). CISH expression at baseline performed well in predicting positive response for Afatinib according to RNA-seq and IHC score (area under the dose-response curve (AUC) = 0.918 and 0.792, respectively) (Fig. 4c and 4g). Furthermore, the participants in this trial harboring higher CISH expression illustrated better OS and EFS tendency versus lower expression participants according to RNA-seq ($p = 0.140$ and 0.057 , respectively) (Fig. 4d-e), as well as in CISH-IHC score ($p = 0.130$ and 0.110 , respectively) (Fig. 4h-i). The representative IHC images for CISH in each responsive group were shown (Fig. 4j-k). Higher CISH RNA-seq expression group was enriched for TCR signaling pathway and T cell mediated cytotoxicity pathway (Fig. 4l-m), which might account for better therapeutic efficacy of EGFR-TKI.

Fig. 4f-k: as the Fig. 4 shown below.

Figure 4: (b) CISH was highly expressed in baseline tumor samples in RNA-seq expression from tumor responders. (c) CISH RNA-seq expression in baseline samples could predict TKI efficiency. (d) The difference in OS between high and low CISH RNAseq expression. (e) The difference in EFS between high and low CISH RNAseq expression. (f) CISH was highly expressed in baseline tumor samples in IHC expression from tumor responders. (g) CISH IHC expression in baseline samples could predict TKI efficiency. (h) The difference in OS between high and low CISH IHC expression. (i) The difference in EFS between high and low CISH IHC expression. Representative IHC images (5X and 40X) for CISH in responder (j) and non-responder (k) tumors. Higher CISH RNA-seq expression group was enriched for TCR signaling pathway (l) and T cell mediated cytotoxicity pathway (m).

Minor issues

4-Longitudinal data should be presented as such where available with connecting lines to identify treatment-associated changes rather than solely overall differences between groups. Likewise, trends pre/post should be compared between responders and non-responders.

Reply 4): Thank you for your comment. These were very meaningful suggestions indeed.

In fact, we have compared the molecular differences in longitudinal data. However, due to the limitation of sample size and the amount of tumor tissue, only 3 pairs longitudinal data for tumors of responders and 5 pairs for tumors of non-responders are existed to analyze. In this circumstance, although the relevant tendencies of molecular changes induced by target-therapy is consistent with the manifested results (as the Reply Fig. 2 shown), the statistical significance is not significant. Therefore, the results of this part are not shown in the manuscript.

Thank you for your suggestions again, as the explanation in the previous reply (Major issue, Reply2), we will enroll more participants and identify the target-therapy associated change through the matched samples of EGFR mutant patients. These trends of change probably guide the novel treatment in the future. We will be happy to edit the manuscript further based on your meaningful comments.

Reply Fig. 2. Immune cell analysis in responder and non-responder patients with matched longitudinal tumor samples.

5-Differences in endothelial cells and fibroblasts observed by MCPcounter should be elaborated upon/discussed.

Reply 5): Thank you for your comment and we have discussed the differences for endothelial cells and fibroblasts in the results part.

Interestingly, we noticed that endothelial cells and fibroblasts were also high after EGFR-TKI treatment. Previous studies proved that high SELP and VWF expression were observed in tumoral high-endothelial venules⁹ (TU-HEVs) and SELP was associated with vascular normalization and immune infiltration¹⁰. In this trial, high SELP and VWF expression were also observed in post-treatment responder tumor samples (**Extended Fig. 2g-h**) and the main components of TU-HEV in post-treatment endothelial cells may attribute to high immune infiltration after EGFR-TKI treatment. Fibroblast was a heterogeneous population, a previous study revealed ADH1B+ CAF mainly come from the CCL19-expressing ADH1B+ cells specifically found in TLS¹¹ in lung cancer. High ADH1B expression (**Extended Fig.2i**) but low FAP expression (**Extended Fig.2j**) after EGFR-TKI treatment might indicated the main contribution of ADH1B+ CAF to fibroblasts, which may also enhance immune infiltration.

Thank you for your comments and we have supplied the concerning contents in the revision manuscript in the red font:

Line 224-230, Page 13-14: After neoadjuvant treatment, high SELP and VWF expression in tumor was observed in responders (Extended Fig. 2g-h), and the main components of TU-HEV endothelial cells may attribute to high immune infiltration. Fibroblast was a heterogeneous population and a previous study revealed ADH1B+ CAF mainly come from the CCL19-expressing ADH1B+ cells specifically found in TLS in lung cancer. High ADH1B (Extended Fig. 2i) but low FAP (Extended Fig. 2j) expression after neoadjuvant treatment might indicate the main contribution of ADH1B+ CAF to fibroblasts, which may enhance immune activity.

Extended Fig. 2g-j: as the Figure shown below.

Extended Fig. 2g-j: High SELP (g), VWF (h) and ADH1B (i) but low FAP (j) RNA-seq expression in post-treatment responder tumor samples.

6-Inclusion of both relative and absolute immune cell quantifications would be informative.

Reply 6): Thank you for your comment and we have applied an absolute immune cell quantification algorithm, EPIC, in our study. Relevant analysis results were shown in the manuscript. For example, we identified high CD8+ T cells and endothelial cells in response tumor samples after TKI via MCPcounter (Fig.5c). By using EPIC, we evaluated the absolute infiltration for immune cells. We noticed that CD8+ T cells and endothelial cells (Fig.5d-e) were also high in response tumor samples after TKI via EPIC.

Fig. 5c-e: as the Figure shown below.

Fig. 5c-e: For tumor responders, (c) Increasing CD8+ T cells and cytotoxic lymphocytes after TKI in tumor samples via MCPcounter. Increasing CD8+ T cells (d) and endothelial cells (e) after TKI in tumor samples via EPIC.

7-Line 113: a $p=0.217$ is generally considered a “better EFS tendency”, this language should be toned down.

Reply 7): Thank you for your comment.

According to your meaningful suggestions and the guidance of statistical experts, we modified the statistical method to evaluate the relationship between EFS and the parameters that should be re-evaluated after neoadjuvant treatment was assessed by the time-dependent Cox model. Surgical treatment, pathological response, and objective response should be re-evaluated after neoadjuvant therapy. Interestingly, in sum of targeted lesions reduction after EGFR-TKI treatment could predict EFS. For the proportion of radiological reduction in sum of lesion diameter (SLD) increasing 1%, the risk of an event occurred decreased by 4.6%.

The concerning contents have been supplemented into the revision manuscript in the red font:

Line 122-126, Page 9: According to the evaluation of the time-dependent Cox model, the proportion of radiological reduction in sum of lesion diameter (SLD) could significantly improve EFS of participants received neoadjuvant Afatinib treatment (HR: 0.954, 95% CI: 0.924-0.985, $p = 0.004$). However, the tendency of EFS could not be observed due to pathological response and surgical treatment (Supplementary Table 3).

8-Considering the impact of Afatinib on T cells – what are the effects on IFN-g pathway, MHC I and MHC II expression and associated chaperones?

Reply 8): Thank you for your comment and we have conducted analysis for IFN pathway and MHCI-II expression.

Results revealed that HLA-E and HLA-DRA showed increasing expression after EGFR-TKI treatment in tumors of responders (**Fig. 5g**). Besides, IFNG pathway showed significant enrichment after EGFR-TKI treatment (**Extended Fig. 2e**). The concerning contents have been supplemented into the revision manuscript in the red font:

Line 216-221, Page 13: In addition, we observed CD8+ T cells (**Fig. 5c**) showed an increasing tendency after treatment along with multiple T cell-related genes, including CD8A expression (**Fig. 5f**). The impact of Afatinib treatment on IFN pathway and MHC family members was also analyzed. The increasing expression of HLA-E/HLA-DRA (**Fig. 5g**) and significant enrichment of IFNG pathway after Afatinib treatment were observed in responder tumor samples (**Extended Fig. 2f**), which may account for the increased T cell infiltration after neoadjuvant treatment.

Fig. 5f-i and Extended Fig. 2f: were shown below.

Fig. 5 f-i: For tumor responders, (f) Increasing T cell related markers after TKI in tumor samples. (g) Increasing MHC members after TKI in tumor samples. (h) Increasing fractions of TCR reads and (i) Unique TCR CDR3 after TKI in tumor samples.

Extended Fig. 2 c-j: (f) Significant enrichment of IFNG pathway in post-treatment responder tumor samples.

9-In Extended Fig. 2F, it is unclear how you can say T cell activity rather than solely infiltration is higher in responders.

Reply 9): Thank you for your suggestion and we have modified our statement in the manuscript. In addition, we replaced the previous CD8+ T cell with CD4+ T cell based on the results from “mcpcounter” and “epic”, as you kindly suggested, which render our results more robust. The concerning contents have been supplemented into the revision manuscript in the red font:

Line 240-241, Page 13: Specifically, the infiltration of CD4+ T cells was also significantly higher in responder tumors (Extended Fig. 4b).

Extended Fig. 4b: were shown below in Extended Fig. 4.

Extended Fig. 4b: (b) Increasing CD4+ T cells infiltration in post-treatment responder tumors via EPIC.

10-A number of axes are either unlabelled (such as Fig. 4) or have unclear axis labels such as “Activity”. Please include units and clarify where applicable.

Reply 10): Thank you for your comment and we have modified labels in the figures (shown below).

Fig. 4 was shown below.

11-Why are changes in MHC II but not MHC I observed according to Extended Fig. 2B?

Reply 11): Thank you for your comment and we have discussed the differences between MHC I and MHC II in the manuscript.

In the analysis between baseline and post-treatment response LN samples, we observed high MHC II expression after EGFR-TKI treatment rather than MHC I members. Along with the high B cell infiltration after EGFR-TKI treatment, the activation of antigen presentation pathway¹² may account for the enhanced TKI efficiency. We have supplemented the concerning contents into the part of Discussion in revision manuscript in the red font:

Line 307-309, Page 17: Along with the high B cell infiltration, high expression of MHC II members after NAT might indicate the activation of antigen presentation pathway in B cell, which may account for the enhanced TKI efficiency in these patients.

Thank you for your comments again, I hope my explanations might answer at least a part of your doubts. We will be happy to edit the manuscript further based on your meaningful comments.

12-Please include PD-L1 status where available and assess impact on clinical and pathological responses.

Reply 12): Thank you for your comment.

We have performed IHC staining for PD-L1 in pre-treatment and post-treatment tumor samples and evaluated its expression by IHC-score (Methods, Immunohistochemistry Analysis), and we have supplemented the concerning contents into the part of Discussion in revision manuscript in the red font:

Line: 439-446, Page: 23: Tumor samples were fixed with formalin and embedded in paraffin. The tumor samples were provided by the Pathology Department of Shanghai Pulmonary Hospital. Stain the slide with Anti-CISH/CIS antibody-C-terminal (Abcam: ab191447) and Anti-PD-L1 antibody (PD-L1: E1L3, Rabbit mAb, CST: 13684S). The immunohistochemical (IHC) score was calculated according to the proportion of positive tumor cells (the proportion less than 1% was 1; the proportion from 1%-49% was 2; the proportion larger than 50% was 3) and the average intensity of positive staining (negative staining was 1; weak staining was 2; and strong staining was 3). The range of IHC score was from 1 to 9 for each sample.

The changes of expression of PD-L1 in tumor samples before and after EGFR-TKI treatment were compared. Besides, the association between PD-L1 and TKI responses was evaluated either. We have supplemented the concerning contents into the part of Results in revision manuscript in the red font:

Line: 203-207, Page: 12: Furthermore, compared with non-responsive tumor, PD-L1 was highly expressed in responsive tumor samples at baseline by IHC score ($p = 0.170$) (Extended Fig. 2c). In addition, compared with pre-treatment tumor samples, significantly higher expression of PD-L1 in primary tumors were observed in post-treatment tumor samples by IHC score ($p = 0.013$) (Extended Fig. 2d), while, this difference was not observed at the transcriptome level ($p = 0.800$) (Extended Fig. 2e).

Extended Fig. 2c-e was shown below.

Extended Fig. 2c-e: (c) High PD-L1 IHC expression tendency in baseline responder tumor samples than non-responder tumor samples. (d) Significantly high PD-L1 IHC expression in post-treatment tumor samples than baseline tumor samples. (e) High PD-L1 RNA-seq expression tendency in post-treatment tumor samples.

Besides, patients with PR to TKI showed high PD-L1 IHC expression tendency in baseline tumor samples ($P = 0.17$) (Extended Fig. 2c) and PD-L1 baseline IHC expression could to some extent predict TKI efficiency ($AUC = 0.667$) (as the Reply Fig. 3 shown below). The expression of PD-L1 in primary tumor does not serve as a routine predictive biomarker for first-line treatment of *EGFR*-mutant NSCLC, so we did not include these results in the revised

manuscript. While, we would compare the ROC curve between PD-L1 and CISH in the next revised version if you insist it is necessary to include it in the manuscript.

Reply Fig. 3. The predictive value of PD-L1 IHC score for TKI response.

Thank you for your suggestions. We will be happy to edit the manuscript further based on your meaningful comments.

13-p-values are missing for certain plots, such as Fig. 4D/E/H/I.

Reply 13): Thank you for your comment and we have made relevant modifications in the figures.

The concerning contents in the last version were demonstrated in Fig. 4d/e/p/q in this reversion manuscript. Thank you for your careful review to make the content of this article more accurate.

Fig. 4d/e/p/q was shown below.

Figure 4. (d) The difference in OS between high and low CISH RNA-seq expression. (e) The difference in EFS between high and low CISH RNA-seq expression. (p) The difference in OS between high and low CISH expression in TCGA-LUAD with *EGFR* mutations. (q) The difference in PFS between high and low CISH expression in TCGA-LUAD with *EGFR* mutations.

14-What role did adjuvant therapies (chemo-IO vs targeted) play in clinical outcomes? This should be elaborated upon.

Reply 14): Thank you for your comments. I'm sorry to mislead you because the contents of the original manuscript are unclear. We have revised this part according to your comments.

In this trial, 33 (70.2%, 33/47) patients received neoadjuvant Afatinib followed by surgical treatment. These patients received adjuvant treatment including target-therapy (Afatinib: 28 Pts, Osimertinib: 2 Pts), and immunochemotherapy (3 Pts). The proportion of patients received adjuvant target-therapy was 90.9% (30/33). Besides, the reasons for choosing adjuvant immunochemotherapy instead of target-therapy as following:

1. BI-019 (male, smoker, Ex19-Del) had stable disease after neoadjuvant treatment. After surgical resection, the results of genomic testing suggested that the EGFR mutant abundance of resected tumor tissue was extremely low. After multidisciplinary discussion, it was considered that target-treatment may not provide survival benefits for him, and adjuvant immunochemotherapy was performed.
2. BI-028 (male, non-smoker, Ex19-Del) had stable disease after neoadjuvant treatment. After surgical resection, the pathological results suggested that the patient had double primary tumors, one of which harboring EGFR mutation and the other with KRAS mutation. EGFR mutant tumor achieved MPR after neoadjuvant treatment, while KRAS mutant tumor did not occur response. After multidisciplinary discussion, adjuvant immunochemotherapy was performed.
3. BI-042 (male, smoker, Ex19-Del) had progressive disease after neoadjuvant treatment. After surgical resection, the pathological results suggested that no novel mutant signal (Ex20-T790M) was found, which was considered as primary Afatinib resistance. After multidisciplinary discussion, adjuvant immunochemotherapy was performed.

We have supplemented the concerning contents into revision manuscript in the red font:

Line 111-116, Page 8: All 33 surgical treated participants received adjuvant treatment postoperatively, including target-therapy (n=30), and immunochemotherapy (n=3). The reasons for performing adjuvant immunochemotherapy are as following: 1) the resected tumor tissue of one participant demonstrated extremely low EGFR mutant abundance; 2) one

participant was diagnosed as double primary tumors harboring *EGFR* and *KRAS* mutation respectively postoperatively; 3) one participant was considered as primary Afatinib resistance.

Fig. 1 was shown below.

Fig. 1: The clinical trial (TEAM-LungMate 004) design

Thank you for your comments again, I hope my explanations might answer at least a part of your doubts.

15-It is unclear what the difference is between the data presented in Fig. 5C and Extended Fig. 2A.

Reply 15): Thank you for your comment.

In fact, Fig. 5C showed the immune cell differences between pre- and post-treatment in response primary tumor samples. Extended Fig. 2A (**Extended Fig. 3A** in this revised version) shows the immune cell differences between pre- and post- treatment in response lymph node samples. Notably, the tumor microenvironment induced by *EGFR*-TKI between primary tumors and involved lymph nodes demonstrated significant difference.

Thank you for your comments again, I hope my explanations might answer at least a part of your doubts.

16-The proposed mechanism accounting for increased T cell infiltration with neoadjuvant Afatinib should be elaborated upon. What about expression of IFN pathway genes, CXCL9/10/11, MHC chaperones, etc?

Reply 16): Thank you for your comment and we have conducted analysis for IFNG pathway, CXCL9/10/11, and MHC expression. Results revealed that HLA-E and HLA-DRA showed increasing expression after EGFR-TKI treatment in response tumor samples (Fig. 5g). IFNG pathway showed significant enrichment after EGFR-TKI treatment (Extended Fig. 2e), which may account for the increased T cell infiltration after neoadjuvant Afatinib treatment. The concerning contents have been supplemented into the revision manuscript in the red font:

Line 218-221, Page 13: The increasing expression of HLA-E/HLA-DRA (Fig. 5g) and significant enrichment of IFNG pathway after Afatinib treatment were observed in responder tumor samples (Extended Fig. 2f), which may account for the increased T cell infiltration after neoadjuvant Afatinib.

Fig. 5 g and Extended Fig. 2f were shown as below.

Fig. 5 f-i: For tumor responders, (f) Increasing T cell related markers after TKI in tumor samples. (g) Increasing MHC members after TKI in tumor samples. (h) Increasing fractions of TCR reads and (i) Unique TCR CDR3 after TKI in tumor samples.

Extended Fig. 2c-j: (f) Significant enrichment of IFNG pathway in post-treatment responder tumor samples.

However, according to the analyses based on RNA-seq, CXCL9/10/11 didn't present with significant changes after Afatinib treatment (as the Reply Fig.4 shown below). Therefore, considering the length limitation of the article, we did not include this section into manuscript.

Reply Fig. 4. CXCL9, CXCL10 and CXCL11 expression levels between baseline and post-treatment tumor samples for responder patients.

Thank you for your comments again, I hope my explanations might answer at least a part of your doubts.

Reviewer #3: with expertise in lung cancer

There are multiple treatment strategies in L/A NSCLC with activating EGFR mutation depending on the tumor size, LN, or operability as follows: 1> upfront surgery ◊ postoperative adjuvant platinum-based chemotherapy followed by adjuvant osimertinib, 2> neoadjuvant chemotherapy followed by surgery, 3>definitive concurrent chemo radiation therapy, 4> upfront EGFR-TKI.

The author conducted neoadjuvant Afatinib phase II trial with 47 stage III NSCLC to investigate the its efficacy in neoadjuvant setting.

Reply: Thank you for you review of this study and kind suggestions. Your suggestions will be of great help to refine this study. What is more, we have also revised the concerning contents in the manuscript according to your suggestions, hoping to make this study more concise and rigorous.

This samples size is too small to identify the efficacy of neoadjuvant afatinib in stage III NSCLC, which were composed of multiple stage (including IIIA-IIIC). The treatment strategies of stage III group would not be unified and their prognosis is so heterogeneous.

Reply 1): Thank you for pointing out this issue. As you mentioned, the sample size of this trial is relatively small. Further, the heterogeneity of NSCLC in stage III is relatively high.

Indeed, for NSCLC harboring EGFR mutation in stage IIIB-IIIC, the standard first-line treatment is target-therapy by EGFR-TKI (including EGFR-TKIs in first-generation (Gefitinib, Erlotinib), second-generation (Afatinib, Dacomitinib), and third generation (Osimertinib)). Besides, As previous clinical trials demonstrated that, EGFR-TKI (Erlotinib¹³ and Osimertinib¹⁴) for stage IIIA EGFR-mutant NSCLC patients is efficacy and safety. Compared with the efficacy of traditional neoadjuvant chemotherapy, neoadjuvant target-therapy significantly improved the ORR and DFS of patients¹⁵. In this circumstance, a prospective clinical trial was designed to assess the efficacy and safety of the second-generation EGFR-TKI for stage III EGFR-mutant NSCLC.

We believed that this trial filled the vacancy of the second-generation EGFR-TKIs in the field of neoadjuvant therapy. Importantly, considering the inevitable resistance of target-therapy, the second-generation of EGFR-TKI was selected for neoadjuvant

therapy, which reserved the opportunity for patients to use the third generation of EGFR-TKI after disease progression or recurrence.

Therefore, I think it is unreasonable to evaluate the effect of neoadjuvant afatinib with 47 people in stage III, where these various diseases are consisted.

Reply 2): Thank you for pointing out this issue. We agree with you that the reliable conclusions could indeed be made by enrolled more participants. In this trial, after neoadjuvant target-therapy, the objective response rate (ORR) of participants was 70.2%, as same as patients received neoadjuvant Osimertinib treatment in NEOS ¹⁴. Besides, the ORR under Afatinib is superior than patients received the first-generation EGFR-TKI neoadjuvant treatment. Interestingly, the difference of major pathological response (MPR) rate could not be observed among patients received neoadjuvant EGFR-TKI treatment in each generation.

In fact, the sample size of this trial, as stated in the manuscript, is calculated by strict statistical methods by comparing previous data ^{13, 15}. More importantly, through prospective, single-arm, open-label clinical trial, the safety and efficacy of neoadjuvant therapy were evaluated by many previous trials (including neoadjuvant chemotherapy, neoadjuvant immunochemotherapy, and target-therapy). Meanwhile, the conversion (resectable patients converted from unresectable patients with treatment) rate by induction therapy was assessed by previous pioneer research in the field of neoadjuvant immunochemotherapy ¹⁶, which has been acknowledged by peers. It also gives us sufficient confidence that the design of this trial is reasonable, and the conclusions are reliable.

Fortunately, the discrepancies between the disease in different stage were assessed in the part of exploratory analysis in this trial. Significant differences of tumor microenvironment could be observed between target-therapy responsive/sensitive patients and non-responsive/non-sensitive patients, which also more straightforward to confirm your point. We will be happy to edit the manuscript further based on your meaningful comments.

Identifying the differences in the same disease is also what we hope to continue in the future to provide more precise treatment for patients. Thank you again for your meaningful comments.

Is there any prespecified post op adjuvant chemotherapy in study protocol? I think that various post op adjuvant treatment in this study could be compounding factors to interpret the survival outcome. Especially, this study enrolled EGFR m+ NSCLC patients. Therefore, adjuvant immunotherapy would be not generally used standard adjuvant therapy in these patients. We found two patients treated with adjuvant therapy in these patients.

Reply 3): Thank you for your comments. I'm sorry to mislead you because the contents of the original manuscript are unclear. We have revised this part according to your comments.

Among all enrolled participants in this trial, 46 patients (97.9%, 46/47) received re-evaluation for the possibility of complete resection after neoadjuvant Afatinib treatment. After re-evaluation, unresectable patients (n=11) and resectable but refuse to surgery patients (n=3) received systematic treatments based on clinical guidelines, including target-therapy, radiotherapy, immunochemotherapy, etc. The remaining 33 (70.2%, 33/47) patients received neoadjuvant Afatinib followed by surgical treatment. All these patients received adjuvant treatment including target-therapy (n=30), and immunochemotherapy (n=3). We have supplemented the concerning contents into revision manuscript in the red font:

Line 111-116, Page 8: All 33 surgical treated participants received adjuvant treatment postoperatively, including target-therapy (n=30), and immunochemotherapy (n=3). The reasons for performing adjuvant immunochemotherapy are as following: 1) the resected tumor tissue of one participant demonstrated extremely low EGFR mutant abundance; 2) one participant was diagnosed as double primary tumors harboring EGFR and KRAS mutation respectively postoperatively; 3) one participant was considered as primary Afatinib resistance.

Fig. 1 was shown as Reply Fig. 9 before.

The proportion of patients received adjuvant target-therapy was 90.9% (30/33). For unresectable patients, according to guidelines, Target-therapy, as the first-line treatment, will continue until the lung cancer progression. **It maximizes the consistency of treatment and thus reduces the bias of survival data caused by the different modes of treatment.** We thought the prognosis in this circumstance in this trial is relatively reliable.

The reasons for choosing adjuvant immunochemotherapy instead of target-therapy as follows:

1. BI-019 (male, smoker, Ex19-Del) had stable disease after neoadjuvant treatment. After surgical resection, the pathological results suggested that the EGFR mutant abundance was extremely low. After multidisciplinary discussion, it was considered that target-treatment may not provide survival benefits for him, and adjuvant immunochemotherapy was performed.
2. BI-028 (male, non-smoker, Ex19-Del) had stable disease after neoadjuvant treatment. After surgical resection, the pathological results suggested that the patient had double primary tumors, one of which harbored EGFR mutation and the other harbored KRAS mutation. EGFR mutant tumor achieved MPR after neoadjuvant treatment, while KRAS mutant tumor did not respond. After multidisciplinary discussion, adjuvant immunochemotherapy was performed.
3. BI-042 (male, smoker, Ex19-Del) had progressive disease after neoadjuvant treatment. After surgical resection, the pathological results suggested that no novel mutant signal (Ex20-T790M) was found, which was considered as primary Afatinib resistance. After multidisciplinary discussion, adjuvant immunochemotherapy was performed.

Thank you for your comments again, I hope my explanations might answer at least a part of your doubts.

Of potentially resectable patients (n=25), approximately half of patients (n=11) did not receive the surgery. The fact that potentially resectable patients did not undergo surgery in half of the patients would be interpreted as not satisfying the goal of neoadjuvant therapy. Or, I think that the initial evaluation of resectability in these patients was not appropriately evaluated.

Reply 4): Thank you for your comments. I'm sorry to mislead you because the contents of the original manuscript are unclear. Indeed, the definition of resectability in NSCLC with stage III disease based on different disease states as well as physician experience is unclear. This is also a subject that attracts us to continue to research. Prospective clinical trial has been conducted at our center to explore the definition of resectable patient after neoadjuvant treatment.

In this trial, resectability was determined by a combination of patient TNM stage, tumor location, lymph node metastasis status, and general physical status. "Potentially Resectable" in this trial means that patients are ineligible for radical resection before neoadjuvant therapy,

possibly because of the site of tumor, the largest diameter of the tumor, or the metastatic status of the mediastinal lymph nodes. Therefore, we thought that resectable patients converted from unresectable patients with neoadjuvant treatment probably provided survival benefit for these patients. **In other words, perhaps the significance of the target-therapy should not be evaluated in terms of the conversion rate of resectability.**

Indeed, for patients who have been evaluated as resectable disease at the time of diagnosis, neoadjuvant therapy does not increase the risk of disease progression, does not elevate the risk of surgery, and improves the survival of patients should be considered as the standard for evaluating the effectiveness of neoadjuvant therapy. In the same way, for potentially resectable (patients with unresectable NSCLC at diagnosis in stage III disease), unresectability after neoadjuvant treatment cannot be used as evidences for poor outcome of neoadjuvant treatment.

Thank you for your comments again, I hope my explanations might answer at least a part of your doubts.

The patients achieved mPR or pCR based on the pathology response after neoadjuvant shows excellent survival outcomes. This is generally accepted concept. I suggest the additional analysis in these patients according to the pathology response.

Reply 5): Thank you for your comments.

Based on the existing data and concepts, ORR is considered to be the first choice of the primary endpoint of neoadjuvant target-therapy for efficacy evaluation. The primary endpoint of prospective clinical trials^{13, 14, 15, 17} (the first/third-generation *EGFR*-TKI) was ORR. Interestingly, according to the time-dependent Cox model, the significant correlation between the EFS and the proportion of sum of lesion diameter (SLD) reduction after neoadjuvant therapy was observed in this trial. To some extent, SLD reduction might replace ORR. ORR might be utilized as a novel predictor of EFS after neoadjuvant second-generation *EGFR*-TKI treatment. We have supplemented the concerning contents into the part of Discussion in revision manuscript in the red font:

Line 268-271, Page 15: Interestingly, according to the time-dependent Cox model, the significant correlation between the EFS and the proportion of SLD reduction after neoadjuvant

therapy was observed. ORR might be utilized as a novel predictor of EFS after neoadjuvant second-generation *EGFR*-TKI treatment.

Line 280-287, Page 16: According to the conclusion of CheckMate-816, neoadjuvant treatment has been recommended as the standard therapy for locally advanced NSCLC. Besides, MPR has been proved as the most crucial prognostic factor for patients. According to the results of previous researches and this study on the treatment of NSCLCm+ with neoadjuvant *EGFR*-TKI, no significant difference was observed in MPR of patients after the neoadjuvant target-therapy with different *EGFR*-TKIs, which is probably related to the fact that *EGFR*-TKI has a predominant cytostatic activity rather than cytotoxicity. As an inhibitor of lung cancer, *EGFR*-TKI has a limited role in curing NSCLCm+. In this circumstance, improving the prognosis of NSCLCm+, especially OS, is an important content of the next phase of research.

Besides, we recognized that a phase 3 prospective clinical trial NeoADAURA¹⁸ choose MPR as the primary endpoint. However, according to the previously published research and the published data of NeoADAURA, the MPR of different neoadjuvant *EGFR*-TKI treatment has no significant difference (approximately 10% to 15%). In this case, the design of prospective neoadjuvant *EGFR*-TKI treatment clinical trials, including this study, cannot take MPR as the primary endpoint, because the selection of MPR will make the sample size of trial that faces higher challenges. Fortunately, in combination with previous data and the results of this study, we will use more reasonable endpoint (such as EFS or DFS) in the next work, so as to make the results of neoadjuvant target-therapy more credible.

We thank you for your suggestions on this issue, and are happy to be able to communicate with you and discuss this crucial issue in the field of target-therapy of *EGFR* mutant NSCLC. We hope that through our communication, we can promote the development of target-therapy and solve the problem of the resistance of *EGFR*-TKIs as soon as possible, in order to find reasonable indicators to predict the treatment efficacy and the prognosis of patients.

Thank you again for your comments.

Reviewer #4: with expertise in lung cancer

Dear Dr Bian and colleagues, it was a great pleasure to review this interesting study which propose the utility of neoadjuvant afatinib in stage III EGFR mutant non-small cell lung cancer.

Reply: Thank you for your review of this study and kind suggestions. Your suggestions will be of great help to refine this study. What is more, we have also revised the concerning contents in the manuscript according to your suggestions, hoping to make this study more concise and rigorous.

Major comments

1) Historically we know that TKIs had a predominant cytostatic activity rather than cytotoxic, and previous works have suggested this kind of therapy just delayed disease recurrence but not cured patients, which is one of the main objectives in this context. I consider adding something about this to your discussion. ¿Would you consider adding chemotherapy in combination with TKIs in this context? Why don't you comment anything about the current phase 3 trial with osimertinib, which is evaluating this issue? What about the increased risk of micrometastatic disease at SNC in stage III patients and the increased risk of recurrence at this site?

Reply 1): Thank you for your comments, as you mentioned, EGFR-TKI is an inhibitor that inhibits the activity of lung cancer, and cannot play the role of radical treatment as chemotherapy and immunotherapy.

As the previous research reported that, compared with chemotherapy, the first-generation EGFR-TKIs (Erlotinib and Gefitinib) could improve PFS^{19, 20, 21, 22}, but could not improve OS of advanced stage patients than chemotherapy treatment (OPTIMAL/CTONG0802)²². Besides, for stage II-III EGFR mutant NSCLC patients, radical surgery followed by adjuvant target-therapy could significantly improve patients DFS versus those who received radical surgery followed by adjuvant chemotherapy. However, no significant difference of OS was observed between the participants in these 2 therapeutic regimens (ADJUVANT/CTONG1104)^{23, 24}. These conclusions demonstrated that **the first-generation EGFR-TKIs only can delay disease recurrence or progression of EGFR mutant NSCLC patients but could not cured them**. According to your suggestion, we supplemented this content into the revision of manuscript (Discussion) in the red font:

Line 250-253, Page 14-15: In addition, the prospective clinical trials (NEOS and NeoADAURA) demonstrated that neoadjuvant third-generation EGFR-TKI (Osimertinib) treatment for locally advanced NSCLCm+ could provide better ORR (70% approximately) than first-generation TKI.

Encouragingly, OS of advanced stage NSCLC harboring Ex19DEL could be improved by Afatinib (the second-generation EGFR-TKI) than these who received chemotherapy as the first-line treatment (LUX-LUNG3 and LUX-LUNG6)²⁵. The conclusions gave us confidence, which apply Afatinib as the first-line treatment for EGFR mutant NSCLC patients.

In recent years, **several clinical trials compared chemotherapy in combination with EGFR-TKI to EGFR-TKI alone**. A phase 3 trial (FASTACT-2)²⁶ demonstrated that combination therapy could improve PFS (16.8 months vs 6.9 months, HR: 0.25, 95% CI: 0.16-0.39, P<0.0001) and OS (31.4 months vs 20.6 months, HR: 0.48, 95% CI: 0.27-0.84, P=0.0092) significantly for EGFR mutant NSCLC patients. For the first-generation EGFR-TKI, combination treatment could not improve PFS (13.7 months vs 10.3 months, HR: 0.62, 95% CI: 0.25-1.57) and OS (37.1 months vs 17.2 months, HR: 0.58, 95% CI: 0.22-1.41) of patients significantly versus TKI monotherapy²⁷. The efficacy of third-generation EGFR-TKI (Osimertinib) combined chemotherapy in the first-line treatment of EGFR mutant NSCLC patients were assessed by OPAL Study²⁸. The ORR of combination treatment was 90.0%, and the PFS was 31.0 months, these 2 parameters were higher than patients received Osimertinib as first-line treatment which was reported in FLAURA²⁹ (PFS: 18.9 months, ORR: 80%). In addition, ADAURA³⁰ evaluated the efficacy of Osimertinib in adjuvant treatment for radical resected EGFR mutant NSCLC in stage II to IIIA. Because of the extremely strong effect of inhibiting the recurrence postoperatively, Osimertinib monotherapy has been recommended as the first choice of adjuvant treatment for stage IB to IIIA patients harboring EGFR mutation.

However, it has not been able to explore appropriate combination therapy to improve the first-line efficacy. If the final OS is negative concerning combination therapy, it may bring the benefits of the second-line to the first-line. chemotherapy is utilized in the first-line or second-line treatment may not affect the final treatment outcome of EGFR-mutant NSCLC patients. According to this circumstance, Osimertinib monotherapy has been recommended as the first-

line treatment for advanced stage EGFR mutant NSCLC, but also as the first choice for adjuvant treatment for complete resected locally advanced EGFR mutant NSCLC.

For locally advanced NSCLC without drive genes mutation, according to the conclusion of CheckMate-816¹, neoadjuvant immunotherapy combined with chemotherapy followed by radical surgery has been recommended as the standard therapeutic regimen. However, the standard neoadjuvant treatment for locally advanced NSCLC harboring EGFR mutation is chemotherapy followed by surgery or definitive concurrent chemoradiation therapy. As we all known, chemotherapy has limited effect on NSCLC, especially for NSCLC harboring EGFR mutation. As a phase 2 clinical trial EMERGING-CTONG1103¹³ demonstrated that the ORR of chemotherapy for EGFR mutant NSCLC was 34.3%. In other words, in clinical work, the surgical conversion (unresectable patients turned into resectable patients) rate of neoadjuvant chemotherapy is limited. EMERGING-CTONG1103 demonstrated that the first-generation EGFR-TKIs could improve the ORR, the rate of complete resection (R0), and DFS significantly compared with chemotherapy for locally advanced EGFR mutant NSCLC. However, as the same as the efficacy of EGFR-TKI for advanced NSCLC, EGFR-TKI could not prolong the OS of patients. However, considering that complete resection is an independent predictor of the prognosis of stage III patients, it is a reasonable choice to give priority to drugs with high surgical conversion rate in neoadjuvant phase³¹. In this circumstance, neoadjuvant Osimertinib for locally advanced NSCLC harboring EGFR mutation has been introduced. As prospective clinical trials (NEOS¹⁴ and NeoADAURA¹⁸) demonstrated that the ORR and MPR was nearly 70% and 10%, respectively. ORR for the third-generation TKI seems much better than the first-generation TKI. Thank you for your comments, and we have supplemented the concerning contents into the revision manuscript in the red font:

Line 280-287, Page 16: According to the conclusion of CheckMate-816, neoadjuvant treatment has been recommended as the standard therapy for locally advanced NSCLC. Besides, MPR has been proved as the most crucial prognostic factor for patients. According to the results of previous researches and this study on the treatment of NSCLCm+ with neoadjuvant EGFR-TKI, no significant difference was observed in MPR of patients after the neoadjuvant target-therapy with different EGFR-TKIs, which is probably related to the fact that EGFR-TKI has a predominant cytostatic activity rather than cytotoxicity. As an inhibitor of lung cancer, EGFR-

TKI has a limited role in curing NSCLCm+. In this circumstance, improving the prognosis of NSCLCm+, especially OS, is an important content of the next phase of research.

Fortunately, these parameters showed consistency with the third-generation TKI in patients receiving Afatinib (the second-generation EGFR-TKI) treatment in our study (TEAM-LungMate004). In addition, considering the inevitable resistance of target-therapy, the second-generation of EGFR-TKI was selected for neoadjuvant therapy, which reserved the opportunity for patients to use the third generation of TKI after disease progression or recurrence. We have supplemented the concerning contents into the revision manuscript in the red font:

Line 126-128, Page 9: Up to the last follow-up, there are 18 participants occurred events, and 11 of them had postoperative recurrence. The proportion of participants with metastasis in central nervous system (CNS) (45.5%, 5/11) is the highest among all recurrent sites.

2) Please I suggest adding something in the discussion about the low rate of major pathologic responses and complete pathologic responses which have been experienced in all EGFR neoadjuvant studies including yours. This could be put in context with this strategy, considering the benefits of other kinds of treatments like immune checkpoint inhibitors. This was in line with my previous comment. Do we really cure more patients with TKIs?

Reply 2): Thank you for your comments. As we all known, the relative low rate of MPR and pCR were observed in nearly all prospective clinical trials and retrospective analysis, whether participants receiving the first-generation or third-generation *EGFR*-TKI.

According to the conclusion of CheckMate-816 ¹, neoadjuvant treatment has been recommended as the standard therapy for locally advanced NSCLC. Besides, **MPR has been proved as the most crucial prognostic factor for patients**. According to the results of previous researches and this study on the treatment of NSCLCm+ with neoadjuvant *EGFR*-TKI, no significant difference was observed in MPR of patients after the neoadjuvant target-therapy with different *EGFR*-TKIs, which is probably related to the fact that *EGFR*-TKI has a predominant cytostatic activity rather than cytotoxicity. As an inhibitor of lung cancer, *EGFR*-TKI has a limited role in curing NSCLCm+. In this circumstance, improving the prognosis of NSCLCm+, especially OS, is an important content of the next phase of research. Thank you

again for your comments, and we have supplemented the concerning contents into the revision manuscript in the red font:

Line 280-287, Page 16: neoadjuvant treatment has been recommended as the standard therapy for locally advanced NSCLC. Besides, MPR has been proved as the most crucial prognostic factor for patients. According to the results of previous researches and this study on the treatment of NSCLCm+ with neoadjuvant EGFR-TKI, no significant difference was observed in MPR of patients after the neoadjuvant target-therapy with different EGFR-TKIs, which is probably related to the fact that EGFR-TKI has a predominant cytostatic activity rather than cytotoxicity. As an inhibitor of lung cancer, EGFR-TKI has a limited role in curing NSCLCm+. In this circumstance, improving the prognosis of NSCLCm+, especially OS, is an important content of the next phase of research.

3) Currently the upfront line treatment in the EGFR metastatic setting is osimertinib due to an increased CNS activity and a more prolonged PFS compared with first-generation TKIs in this population. Furthermore, the main site of recurrence especially in the stage III population is CNS, please justify your decision to opt for a second-generation TKI than a third-generation in your study design. Additionally, why don't you analyze sites of recurrence? It would be interesting to know the proportion of patients who had a CNS recurrence.

Reply 3): Thank you for your comments. These comments will help us make up for the deficiencies of previous analysis and make the conclusions more complete.

As you mentioned, the most common postoperative recurrence site is CNS. In our research, at the cut-off date of May 2023, in all 47 participants, there are 18 patients occurred events. Of these 18 patients, 11 patients had postoperative recurrence, including 5 patients with CNS metastases (5/11, 45.5%). We have supplemented the concerning contents into the revision manuscript (Results, Survival Analysis) in the red font:

Line 126-128, Page 9: Up to the last follow-up, there are 18 participants occurred events, and 11 of them had postoperative recurrence. The proportion of participants with metastasis in central nervous system (CNS) (45.5%, 5/11) is the highest among all recurrent sites.

As previous prospective trial (FLAURA^{32,33}) demonstrated that, for untreated advanced NSCLC patients harboring EGFR mutation, Osimertinib could provide longer OS (38.6 months

vs 31.8 months, HR: 0.80, $P=0.046$) than those who received first *EGFR*-TKIs. In addition, compared with any other treatment, the proportion of adverse events in Osimertinib treated patients is lower. Last but not least, compared with the first-generation *EGFR*-TKI, Osimertinib has a higher efficacy of inhibition on patients with brain metastasis. At the same time, Osimertinib treated patients have a lower proportion of progress due to distant metastasis. These conclusions proved that the third-generation *EGFR*-TKI has better therapeutic efficacy than the first-generation *EGFR*-TKI. Regrettably, there are no relevant prospective trials comparing the efficacy of the second-generation *EGFR*-TKI to the third-generation *EGFR*-TKI. An observational study (GioTag study³⁴) assessed the survival of *EGFR* mutant NSCLC patients who received sequential Afatinib and Osimertinib, which calculated that median OS for patients in this therapeutic regimen was 41.3 months. If patients respond to Osimertinib after resistance to first-line Afatinib treatment, their outcomes may be better than patients treated Osimertinib as the first-line treatment.

Considering these conclusions, **the second-generation of *EGFR*-TKI was selected for neoadjuvant therapy, which reserved the opportunity for patients to use the third-generation *EGFR*-TKI after disease progression or recurrence.** We have supplemented the concerning contents into the part of Discussion in revision manuscript in the red font:

Line 256-258, Page 15: In addition, in view of the mechanism of *EGFR*-TKI resistance is unclear, the results of this trial reserve the opportunity for recurrent patients to continue to use the third-generation *EGFR*-TKI.

4) Bearing in mind that adjuvant therapy is the standard treatment in resectable stage III patients who underwent surgical resection. Why don't you standardize adjuvant therapy in your study? Some patients received *EGFR* TKI, others chemotherapy, but up to 25% didn't receive any post-surgical therapy. How reliable is the EFS endpoint, with so much variability in the adjuvant setting? Please explain, why in this stage III population up to one-fourth of patients did not receive any adjuvant treatment.

Reply 4): Thank you for your comments. I'm sorry to mislead you because the contents of the original manuscript are unclear. We have revised this part according to your comments.

Among all enrolled participants in this trial, 46 patients (97.9%, 46/47) received re-evaluation for the possibility of complete resection after neoadjuvant Afatinib treatment. After re-evaluation, unresectable patients (n=11 pts) and resectable but refuse to surgery patients (n=3 pts) received systematic treatments based on clinical guidelines, including target-therapy, radiotherapy, immunochemotherapy, etc. The remaining 33 (70.2%, 33/47) patients received neoadjuvant Afatinib followed by surgical treatment. All these patients received adjuvant treatment including target-therapy (Afatinib: n=28 pts, Osimertinib: n=2 pts), and immunochemotherapy (n=3 pts). We have supplemented the concerning contents into revision manuscript (**Figure 1**) in the red font:

Line 111-116, Page 8: All 33 surgical treated participants received adjuvant treatment postoperatively, including target-therapy (n=30), and immunochemotherapy (n=3). The reasons for performing adjuvant immunochemotherapy are as following: 1) the resected tumor tissue of one participant demonstrated extremely low EGFR mutant abundance; 2) one participant was diagnosed as double primary tumors harboring EGFR and KRAS mutation respectively postoperatively; 3) one participant was considered as primary Afatinib resistance.

The proportion of patients received adjuvant target-therapy was 90.9% (30/33). For unresectable patients, according to guidelines, Target-therapy, as the first-line treatment, will continue until the lung cancer progression. We thought the EFS in this circumstance is relatively reliable.

The reasons for choosing adjuvant immunochemotherapy instead of target-therapy as follows:

1. BI-019 (male, smoker, Ex19-Del) had stable disease after neoadjuvant treatment. After surgical resection, the pathological results suggested that the EGFR mutant abundance was extremely low. After multidisciplinary discussion, it was considered that target-treatment may not provide survival benefits for him, and adjuvant immunochemotherapy was performed.
2. BI-028 (male, non-smoker, Ex19-Del) had stable disease after neoadjuvant treatment. After surgical resection, the pathological results suggested that the patient had double primary tumors, one of which harbored EGFR mutation and the other harbored KRAS mutation. EGFR mutant tumor achieved MPR after neoadjuvant treatment, while KRAS

mutant tumor did not response. After multidisciplinary discussion, adjuvant immunochemotherapy was performed.

3. BI-042 (male, smoker, Ex19-Del) had progressive disease after neoadjuvant treatment. After surgical resection, the pathological results suggested that no novel mutant signal (Ex20-T790M) was found, which was considered as primary Afatinib resistance. After multidisciplinary discussion, adjuvant immunochemotherapy was performed.

Thank you for your comments again, I hope my explanations might answer at least a part of your doubts.

5) Another key point to be discussed is to clarify, why did you choose the objective response rate as the primary endpoint? As it was stated in the introduction, and it was clear in the CTONG 1103 a non-negligible number of patients do not reach tumor response due to the appearance of necrosis which can sub-estimate the downstaging of tumors. Thus, many neoadjuvant studies choose for more solid outcomes like major pathologic response, which in some contexts has been demonstrated to be a surrogate of survival.

Reply 5): Thank you for your comments.

Based on the existing data and concepts, ORR is considered to be the first choice of the primary endpoint of neoadjuvant target-therapy for efficacy evaluation. The primary endpoint of prospective clinical trials^{13, 14, 15, 17} (the first/third-generation *EGFR*-TKI) was ORR. Interestingly, according to the time-dependent Cox model, the significant correlation between the EFS and the proportion of sum of lesion diameter (SLD) reduction after neoadjuvant therapy was observed in this trial. **ORR might be utilized as a novel predictor of EFS after neoadjuvant second-generation *EGFR*-TKI treatment.** We have supplemented the concerning contents into the part of Results and Discussion in revision manuscript in the red font:

Line 122-126, Page 9: According to the evaluation of the time-dependent Cox model, the proportion of radiological reduction in sum of lesion diameter (SLD) could significantly improve EFS of participants received neoadjuvant Afatinib treatment (HR: 0.954, 95% CI: 0.924-0.985, $p = 0.004$). However, the tendency of EFS could not be observed due to pathological response and surgical treatment (Supplementary Table 3).

Line 268-271, Page 15: Interestingly, according to the time-dependent Cox model, the significant correlation between the EFS and the proportion of SLD reduction after neoadjuvant therapy was observed. ORR might be utilized as a novel predictor of EFS after neoadjuvant second-generation *EGFR*-TKI treatment.

Besides, we recognized that a phase 3 prospective clinical trial NeoADAURA¹⁸ choose MPR as the primary endpoint. However, according to the previously published research and the published data of NeoADAURA, the MPR of different neoadjuvant *EGFR*-TKI treatment has no significant difference (approximately 10% to 15%). In this case, the design of prospective neoadjuvant *EGFR*-TKI treatment clinical trials, including this study, cannot take MPR as the primary endpoint, because the selection of MPR will make the sample size of trial that faces higher challenges. Fortunately, in combination with previous data and the results of this study, we will use more reasonable endpoint (such as EFS or DFS) in the next work, so as to make the results of neoadjuvant target-therapy more credible.

We thank you for your suggestions on this issue, and are happy to be able to communicate with you and discuss this crucial issue in the field of target-therapy of *EGFR* mutant NSCLC. We hope that through our communication, we can promote the development of target-therapy and solve the problem of the resistance of *EGFR*-TKIs as soon as possible, in order to find reasonable indicators to predict the treatment efficacy and the prognosis of patients.

Thank you again for your comments.

6) I like so much the analysis of tumor microenvironment and molecular gene analysis. In this regard, I have some recommendations.

- It will be interesting if it is possible to add an analysis of PD-L1 expression before and after NAT.

- Second, it was quite interesting the increase in VEGF-B expression after NAT. I suggest discussing these results deeply. Based on these results, would you consider adding antiangiogenics in this context? This analysis opens the window to assess this combination strategy (*EGFR* TKI with antiangiogenic) to reach better responses.

Reply 6): We thank you for your overall positive attitude towards our bioinformatic analysis.

We performed IHC staining for PD-L1 in baseline and post-treatment tumor samples and evaluated its expression by IHC-score. Results revealed that PD-L1 showed an increasing expression after treatment in IHC-score (**Extended Fig. 2d**). In addition, we also compared the RNA-seq PD-L1 expression between baseline and post-treatment tumor samples and found that PD-L1 also showed an increasing tendency after NAT (**Extended Fig. 2e**). We have supplemented the concerning contents into the part of Result in revision manuscript in the red font:

Line 196-207, Page 12: The expression of programmed cell death 1 ligand 1 (PD-L1) was assessed by IHC score in 12 and 11 primary tumor samples before and after neoadjuvant Afatinib treatment in this trial respectively. Before treatment (baseline), there were 10 (10/12, 83.3%) and 2 (2/12, 16.7%) tumors with negative and positive expression of PD-L1 respectively. Seven participants (7/11, 63.6%) had positive PD-L1 expression in primary tumor after neoadjuvant Afatinib treatment, including one with high expression. By comparing the changes in 9 paired samples before and after Afatinib treatment, elevated PD-L1 expression after treatment was observed in 3 responders and 2 non-responders (Extended Fig. 2a-b). Furthermore, compared with non-responsive tumor, PD-L1 was highly expressed in responsive tumor samples at baseline by IHC score ($p = 0.170$) (Extended Fig. 2c). In addition, compared with pre-treatment tumor samples, significantly higher expression of PD-L1 in primary tumors were observed in post-treatment tumor samples by IHC score ($p = 0.013$) (Extended Fig. 2d), while, this difference was not observed at the transcriptome level ($p = 0.800$) (Extended Fig. 2e).

Extended Fig. 2a-e was shown below.

Extended Fig. 2a-e: Extended Figure 2: (a) The IHC images (10X) of PD-L1 in pre-treatment and post-treatment tumor samples from responders. (b) The IHC images (10X) of PD-L1 in pre-treatment and post-treatment tumor samples from non-responders. (c) High PD-L1 IHC expression tendency in baseline responder tumor samples than non-responder tumor samples. (d) Significantly high PD-L1 IHC expression in post-treatment tumor samples than baseline tumor samples. (e) High PD-L1 RNA-seq expression tendency in post-treatment tumor samples.

We agreed with reviewer's comment for indication of VEGFB over-expression for combination treatment. VEGFB was a member of the vascular endothelial growth factor

receptor family³⁵ and was involved in embryonic angiogenesis³⁶. Combination treatment between antiangiogenics³⁷ and Afatinib maybe a novel treatment strategy especially for non-responder patients to *EGFR*-TKI monotherapy. We have supplemented the concerning contents into the part of Discussion in revision manuscript in the red font:

Line 324-328, Page 18: In addition, VEGFB showed increased expression in tumor samples of non-responder after treatment. VEGFB was a member of the vascular endothelial growth factor receptor family and was involved in embryonic angiogenesis. Combination treatment between antiangiogenics and Afatinib maybe a potential treatment strategy especially for those non-responder patients.

Thank you again for your comments.

7) In the differential expression analysis, do you assess TP53?

Reply 7): Thank you for your comments. This suggestion has given us new exploration directions for combination treatment and resistance mechanism in the next work. TP53 is a classic tumor suppressor gene and, to our knowledge, patients harboring *EGFR* mutation demonstrated higher rate of mutated type TP53 than *EGFR* wild type patients³⁸. Besides, in advanced stage NSCLC patients with TP53 mutation have poor prognosis and lower response rate to target-therapy and/or immunochemotherapy^{38,39}.

In this circumstance, TP53 expression of participants' tumor samples before and after neoadjuvant Afatinib treatment in this trial was assessed, but no significant difference was observed (as the Reply Fig. 5 shown below).

Reply Fig. 5. TP53 expression level between baseline and post-treatment tumor samples.

Thank you again for your comments and we are also interested in the exploration of therapeutic regimens for co-mutation NSCLC patients harboring EGFR mutation, and will be carried out in the following studies.

8) In the ORR table (table 2), patients with less than 60% of residual disease were greater than 50%. In other studies, this cut-off point of the residual disease has suggested better outcomes. Why does this information not discuss? Please explain, why do you choose a cutoff point of 60%? and what is the relevance of this finding in your study? In this regard, don't you consider it interesting to assess an event-free survival curve stratified by this 60% cut-off point?

Reply 8): Thank you for your comments. Considering the limitations of target-therapy, the researches concerning the MPR and the proportion of residual tumor cell are relatively difficult and limited. Before carrying out the statistical work of the proportion of residual tumor cells after neoadjuvant treatment, we read a lot of previous data and literatures.

As previous researches in the field of lung cancer, many prospective trials have evaluated the correlation between pCR and OS. OS was significantly improved by neoadjuvant chemotherapy in patients with stage III NSCLC who had a pCR⁴⁰. Besides, a retrospective analysis demonstrated that the survival of patients with smaller than 10% residual tumor cells (MPR) had significantly longer survival than other patients (36 months vs 14 months, $P=0.02$). MPR could be considered a surrogate of OS in patients with resectable NSCLC treated with neoadjuvant

chemotherapy/immunochemotherapy^{41, 42}. Because the incidence of pCR and MPR is relatively lower, especially in EGFR-mutant NSCLC, it is necessary to establish a more specific model of pathological response rate/residual tumor cells to predict the curative effect and survival.

Continue to search for evidence, a prospective trial⁴¹ evaluated the degree of pathologic response under neoadjuvant chemotherapy for stage IIIA NSCLC patients. The median pathologic response was 60%, and patients with more than 60% pathologic response had significantly longer median OS than those with <60% response (61 months vs 22 months, $P=0.03$).

Combined with the above evidences and the actual situation of this trial (the median pathologic response is approximately 60%), the cut-off value for assessing pathological response was set at 60%. As recommended by statistical experts, the correlation between pathologic response and EFS was established according to neoadjuvant treatment cycle by the time-dependent Cox model. However, no significant correlation was observed between pathologic response and EFS (HR: 0.974, $P=0.200$). In a sense, establishing a correlation of pathological response and EFS on the basis of continuing adjuvant therapy may require a longer follow-up period. We have supplemented these comments into the revision manuscript (Supplementary Table 3) in the red font:

Line 122-126, Page 9: According to the evaluation of the time-dependent Cox model, the proportion of radiological reduction in sum of lesion diameter (SLD) could significantly improve EFS of participants received neoadjuvant Afatinib treatment (HR: 0.954, 95% CI: 0.924-0.985, $p = 0.004$). However, the tendency of EFS could not be observed due to pathological response and surgical treatment (Supplementary Table 3).

Thank you very much for your suggestions and direction for our subsequent work.

9) Do you assess the percentage of necrosis in post-treatment samples? And if it is the case, do you explore the correlation between necrosis and response?

Reply 9): Thank you for your comments.

We re-evaluated the percentage of necrosis in post-treatment samples. Each pathological biopsy was composed of several components: tumor, necrotic tissue, and stroma. According to evaluation of the samples postoperatively, stroma could be observed in every 33 participants

(median percentage: 30%, range from 1% to 100%). Besides, necrotic tissue was observed in 11 participants (median percentage: 5%, range from 2% to 10%). What is more, no significant correlation was observed between these parameters and efficacy (pathological or radiological).

Thank you again for your comments.

Minor comments

1) Between lines 212-216 is suggested that afatinib had a greater efficacy in terms of response rate than first-generation TKIs. I'm not convinced that this can be supported by the cited evidence and their results. Erlotinib in the CTONG 1103 was administered orally for a median of 42 days with an ORR of 53%. In contrast in your study, afatinib was administered until 4 cycles of treatment with a median of 2.7 and an ORR of 70.2%. Moreover, your exploratory analysis clearly shows that patients who received more than two cycles reach better responses (94% vs 69%), thus, differences between first-EGFR TKIs studies and yours could be explained by the number of cycles administered in the previous surgery.

Reply 1): Thank you for your comments.

We ignored the difference between treatment cycles in previous studies and this study in the last version of the manuscript. Based on this circumstance, we performed a more precise description in the revised version. We could not conclude the superiority between two generations *EGFR*-TKIs by contrast with previous research (CTONG-1103, etc.). But we can really draw the conclusion of earlier anti-oncologic onset of second-generation *EGFR*-TKI that make the cycle of partial response shorter than first-generation *EGFR*-TKI. At the same time, we also concluded that moderate prolongation of induction *EGFR*-TKI therapy might improve the success rates of surgical rate. We have supplemented these comments into the revision manuscript in the red font:

Line: 265-268, Page15: It probably shows that Afatinib has earlier anti-oncologic onset and greater therapeutic effect versus the first-generation *EGFR*-TKIs. The above advantages may support that Afatinib is suitable for neoadjuvant therapy.

2) Please review lines 143-145 on page 10, it's quite hard to understand. I suggest rewriting this sentence.

Reply 2): Thank you for your comments. This paragraph mainly discusses the safety of neoadjuvant treatment followed by surgery. In this trial, after neoadjuvant target-therapy, 4 patients occurred surgical related complications. Fortunately, all of them recovered after non-surgical procedures which confirmed the safety of neoadjuvant Afatinib therapy.

And we will re-write this part in the revision manuscript in the red font as following:

Line: 156-157, Page10: Surgery-related complications occurred in four patients, of whom there and one experienced prolonged air leakage (3/33, 9.1%) and bronchopleural fistula (1/33, 3.0%), respectively.

3) I suggest replacing the phrase "increasing tendency" in most of the paragraphs, my impression is that this phrase is overused in your manuscript. In addition, I understand that statistical differences did not exist in many comparisons, however, I did not consider the most appropriate way to describe a non-significant difference.

Reply 3): Thank you for your comments. According to your suggestion, we re-wrote the relevant contents in the revised version. Thank you for your careful and strict review of this manuscript, which will make it more rigorous.

4) BCR is missing in the abbreviation and acronyms list.

Reply 4): Thank you for your comments.

We supplemented the concerning abbreviation (TCR and BCR) in the revision manuscript (Abbreviations and Acronyms List) in the red font:

Page 14, Line 231; Page 14, Line 235; and Abbreviations and Acronyms List.

Thank you for your careful and strict review of this manuscript, which will make it more rigorous.

5) I suggest in general, improving English. In some fragments of the manuscript is quite difficult to follow the idea.

Reply 5): Thank you for your comments.

We apologize for the poor language of our manuscript. We worked on the manuscript for a long time and the repeated addition and removal of sentences and sections obviously led to

poor readability. We have worked on both language and readability and have also involved native English speakers for language corrections. We really hope that the flow and language level have been substantially improved.

Thank you again for your review and suggestions on this article.

References

1. Forde P, *et al.* Neoadjuvant Nivolumab plus Chemotherapy in Resectable Lung Cancer. *The New England journal of medicine* **386**, 1973-1985 (2022).
2. Offin M, *et al.* Tumor Mutation Burden and Efficacy of EGFR-Tyrosine Kinase Inhibitors in Patients with EGFR-Mutant Lung Cancers. *Clin Cancer Res* **25**, 1063-1069 (2019).
3. Isomoto K, *et al.* Impact of EGFR-TKI Treatment on the Tumor Immune Microenvironment in EGFR Mutation-Positive Non-Small Cell Lung Cancer. *Clin Cancer Res* **26**, 2037-2046 (2020).
4. Hellmann MD, *et al.* Genomic Features of Response to Combination Immunotherapy in Patients with Advanced Non-Small-Cell Lung Cancer. *Cancer Cell* **33**, 843-852.e844 (2018).
5. Liu D, *et al.* Integrative molecular and clinical modeling of clinical outcomes to PD1 blockade in patients with metastatic melanoma. *Nat Med* **25**, 1916-1927 (2019).
6. Samstein RM, *et al.* Tumor mutational load predicts survival after immunotherapy across multiple cancer types. *Nat Genet* **51**, 202-206 (2019).
7. Karn T, *et al.* Tumor mutational burden and immune infiltration as independent predictors of response to neoadjuvant immune checkpoint inhibition in early TNBC in GepardNuevo. *Ann Oncol* **31**, 1216-1222 (2020).
8. Hu J, *et al.* Tumor microenvironment remodeling after neoadjuvant immunotherapy in non-small cell lung cancer revealed by single-cell RNA sequencing. *Genome Med* **15**, 14 (2023).

9. Hua Y, *et al.* Cancer immunotherapies transition endothelial cells into HEVs that generate TCF1(+) T lymphocyte niches through a feed-forward loop. *Cancer Cell* **40**, 1600-1618.e1610 (2022).
10. Elstad MR, *et al.* P-selectin regulates platelet-activating factor synthesis and phagocytosis by monocytes. *J Immunol* **155**, 2109-2122 (1995).
11. Grout JA, *et al.* Spatial Positioning and Matrix Programs of Cancer-Associated Fibroblasts Promote T-cell Exclusion in Human Lung Tumors. *Cancer Discov* **12**, 2606-2625 (2022).
12. Axelrod ML, Cook RS, Johnson DB, Balko JM. Biological Consequences of MHC-II Expression by Tumor Cells in Cancer. *Clin Cancer Res* **25**, 2392-2402 (2019).
13. Zhong W, *et al.* EGFR Erlotinib Versus Gemcitabine Plus Cisplatin as Neoadjuvant Treatment of Stage IIIA-N2 -Mutant Non-Small-Cell Lung Cancer (EMERGING-CTONG 1103): A Randomized Phase II Study. *Journal of clinical oncology : official journal of the American Society of Clinical Oncology* **37**, 2235-2245 (2019).
14. Lv C, *et al.* Osimertinib as neoadjuvant therapy in patients with EGFR-mutant resectable stage II-III B lung adenocarcinoma (NEOS): A multicenter, single-arm, open-label phase 2b trial. *Lung Cancer* **178**, 151-156 (2023).
15. Zhang Y, *et al.* Gefitinib as neoadjuvant therapy for resectable stage II-III A non-small cell lung cancer: A phase II study. *J Thorac Cardiovasc Surg*, (2020).
16. Zhu X, *et al.* Safety and effectiveness of neoadjuvant PD-1 inhibitor (toripalimab) plus chemotherapy in stage II-III NSCLC (LungMate 002): an open-label, single-arm, phase 2 trial. *BMC Med* **20**, 493 (2022).

17. Xiong L, *et al.* Erlotinib as Neoadjuvant Therapy in Stage IIIA (N2) EGFR Mutation-Positive Non-Small Cell Lung Cancer: A Prospective, Single-Arm, Phase II Study. *Oncologist* **24**, 157-e164 (2019).
18. Tsuboi M, *et al.* Neoadjuvant osimertinib with/without chemotherapy versus chemotherapy alone for EGFR-mutated resectable non-small-cell lung cancer: NeoADAURA. *Future Oncol*, (2021).
19. Mitsudomi T, *et al.* Gefitinib versus cisplatin plus docetaxel in patients with non-small-cell lung cancer harbouring mutations of the epidermal growth factor receptor (WJTOG3405): an open label, randomised phase 3 trial. *Lancet Oncol* **11**, 121-128 (2010).
20. Zhou C, *et al.* Erlotinib versus chemotherapy as first-line treatment for patients with advanced EGFR mutation-positive non-small-cell lung cancer (OPTIMAL, CTONG-0802): a multicentre, open-label, randomised, phase 3 study. *Lancet Oncol* **12**, 735-742 (2011).
21. Rosell R, *et al.* Erlotinib versus standard chemotherapy as first-line treatment for European patients with advanced EGFR mutation-positive non-small-cell lung cancer (EURTAC): a multicentre, open-label, randomised phase 3 trial. *The Lancet Oncology* **13**, 239-246 (2012).
22. Zhou C, *et al.* Final overall survival results from a randomised, phase III study of erlotinib versus chemotherapy as first-line treatment of EGFR mutation-positive advanced non-small-cell lung cancer (OPTIMAL, CTONG-0802). *Ann Oncol* **26**, 1877-1883 (2015).

23. Zhong W, *et al.* Gefitinib versus vinorelbine plus cisplatin as adjuvant treatment for stage II-III A (N1-N2) EGFR-mutant NSCLC (ADJUVANT/CTONG1104): a randomised, open-label, phase 3 study. *The Lancet Oncology* **19**, 139-148 (2018).
24. Zhong WZ, *et al.* Gefitinib Versus Vinorelbine Plus Cisplatin as Adjuvant Treatment for Stage II-III A (N1-N2) EGFR-Mutant NSCLC: Final Overall Survival Analysis of CTONG1104 Phase III Trial. *J Clin Oncol* **39**, 713-722 (2021).
25. Yang J, *et al.* Afatinib versus cisplatin-based chemotherapy for EGFR mutation-positive lung adenocarcinoma (LUX-Lung 3 and LUX-Lung 6): analysis of overall survival data from two randomised, phase 3 trials. *The Lancet Oncology* **16**, 141-151 (2015).
26. Wu YL, *et al.* Intercalated combination of chemotherapy and erlotinib for patients with advanced stage non-small-cell lung cancer (FASTACT-2): a randomised, double-blind trial. *Lancet Oncol* **14**, 777-786 (2013).
27. Gijtenbeek RGP, *et al.* Randomised controlled trial of first-line tyrosine-kinase inhibitor (TKI) versus intercalated TKI with chemotherapy for EGFR-mutated nonsmall cell lung cancer. *ERJ Open Res* **8**, (2022).
28. Saito R, *et al.* Phase 2 Study of Osimertinib in Combination with Platinum and Pemetrexed in Patients with Previously Untreated EGFR-Mutated Advanced Non-Squamous Non-Small Cell Lung Cancer: The OPAL Study. *European Journal of Cancer*, (2023).
29. Soria JC, *et al.* Osimertinib in Untreated EGFR-Mutated Advanced Non-Small-Cell Lung Cancer. *N Engl J Med* **378**, 113-125 (2018).

30. Wu YL, *et al.* Osimertinib in Resected EGFR-Mutated Non-Small-Cell Lung Cancer. *N Engl J Med* **383**, 1711-1723 (2020).
31. Xiong Y, *et al.* The efficacy of neoadjuvant EGFR-TKI therapy combined with radical surgery for stage IIIB lung adenocarcinoma harboring EGFR mutations: A retrospective analysis based on single center. *Front Oncol* **13**, 1034897 (2023).
32. Soria JC, *et al.* Osimertinib in Untreated EGFR-Mutated Advanced Non-Small-Cell Lung Cancer. *N Engl J Med* **378**, 113-125 (2018).
33. Ramalingam SS, *et al.* Overall Survival with Osimertinib in Untreated, EGFR-Mutated Advanced NSCLC. *N Engl J Med* **382**, 41-50 (2020).
34. Hochmair MJ, *et al.* Sequential afatinib and osimertinib in patients with EGFR mutation-positive non-small-cell lung cancer: updated analysis of the observational GioTag study. *Future Oncol* **15**, 2905-2914 (2019).
35. Korpanty G, Sullivan LA, Smyth E, Carney DN, Brekken RA. Molecular and clinical aspects of targeting the VEGF pathway in tumors. *J Oncol* **2010**, 652320 (2010).
36. Melincovici CS, *et al.* Vascular endothelial growth factor (VEGF) - key factor in normal and pathological angiogenesis. *Rom J Morphol Embryol* **59**, 455-467 (2018).
37. Le X, *et al.* Dual EGFR-VEGF Pathway Inhibition: A Promising Strategy for Patients With EGFR-Mutant NSCLC. *J Thorac Oncol* **16**, 205-215 (2021).
38. Jiao XD, Qin BD, You P, Cai J, Zang YS. The prognostic value of TP53 and its correlation with EGFR mutation in advanced non-small cell lung cancer, an analysis based on cBioPortal data base. *Lung Cancer* **123**, 70-75 (2018).

39. Labbé C, *et al.* Prognostic and predictive effects of TP53 co-mutation in patients with EGFR-mutated non-small cell lung cancer (NSCLC). *Lung Cancer* **111**, 23-29 (2017).
40. Hellmann MD, *et al.* Pathological response after neoadjuvant chemotherapy in resectable non-small-cell lung cancers: proposal for the use of major pathological response as a surrogate endpoint. *Lancet Oncol* **15**, e42-50 (2014).
41. Betticher DC, *et al.* Mediastinal lymph node clearance after docetaxel-cisplatin neoadjuvant chemotherapy is prognostic of survival in patients with stage IIIA pN2 non-small-cell lung cancer: a multicenter phase II trial. *J Clin Oncol* **21**, 1752-1759 (2003).
42. Chaft JE, *et al.* Phase II trial of neoadjuvant bevacizumab plus chemotherapy and adjuvant bevacizumab in patients with resectable nonsquamous non-small-cell lung cancers. *J Thorac Oncol* **8**, 1084-1090 (2013).

REVIEWERS' COMMENTS

Reviewer #1 (Remarks to the Author):

Thank you for the thorough update. Looks good.

Reviewer #2 (Remarks to the Author):

The authors have done an excellent job of responding to my initial critiques. I believe most of the new analyses add value and should be included in the manuscript at minimum as supplemental figures. A few issues remain.

Initial comment #:

2) Please assess changes in T cell repertoire in matched samples, including Jaccard and Morisita index and track changes in clone frequencies to identify any therapy- or response-associated changes.

10) y-axis labels/units are still missing for b, f, n, o.

Reviewer #3 (Remarks to the Author):

The authors provided thorough responses to the issues I raised in my previous review. However, I still have doubts about the significance of this study, which was conducted on a small sample size of heterogeneous stage III lung cancer patients, as I mentioned during the review. Furthermore, I have concerns about whether this study is suitable for the standards of this journal. I believe that this study may not have the priority to be published in a journal like Nature Communications.

My final decision of this article is "Reject"

Reviewer #4 (Remarks to the Author):

After a deep review I consider the manuscript has improved substantially since the last version. According to my field of expertise, all answers were appropriate and most important those situations which cannot be changed or completed were understandable and limitations were stated at the discussion section.

In my opinion, after some minor changes which I observed still pendant that were pointed out by other reviewers. I consider this paper suitable to be published in this Journal.

Reviewer #1 (Remarks to the Author):

Thank you for the thorough update. Looks good.

Reply: Thank you for your hard and nice work for our manuscript. Your suggestions make our work more precise and rigorous. Thank you again for your approval.

Reviewer #2 (Remarks to the Author):

The authors have done an excellent job of responding to my initial critiques. I believe most of the new analyses add value and should be included in the manuscript at minimum as supplemental figures. A few issues remain.

Initial comment #:

2) Please assess changes in T cell repertoire in matched samples, including Jaccard and Morisita index and track changes in clone frequencies to identify any therapy- or response-associated changes.

10) y-axis labels/units are still missing for b, f, n, o.

Reply: Thank you for your hard and nice work for our manuscript. Your suggestions make our work more precise and rigorous. As you mentioned, we supplement the results, as you mentioned, into the Supplementary Figure 3, which includes the immune cell and repertoire differences in matched samples. Meanwhile, some therapy-associated clones were identified and Jaccard/Morisita indexes were evaluated and presented in Supplementary Figure 3. In addition, some y-axis labels were added as you mentioned. Thank you again for your approval.

Reviewer #3 (Remarks to the Author):

The authors provided thorough responses to the issues I raised in my previous review. However, I still have doubts about the significance of this study, which was conducted on a small sample size of heterogeneous stage III lung cancer patients, as I mentioned during the review. Furthermore, I have concerns about whether this study is suitable for the standards of this journal. I believe that this study may not have the priority to be published in a journal like Nature Communications.

My final decision of this article is "Reject"

Reply: Thank you for your hard and nice work for our manuscript. Your suggestions make our work more precise and rigorous. It's a pity that the revised manuscript still hasn't received your approval, but this article has been revised according to your comments and suggestions, with clearer expression, more accurate results and more rigorous conclusions. Thank you again for your work.

Reviewer #4 (Remarks to the Author):

After a deep review I consider the manuscript has improved substantially since the last version. According to my field of expertise, all answers were appropriate and most important those situations which cannot be changed or completed were understandable and limitations were stated at the discussion section.

In my opinion, after some minor changes which I observed still pendant that were pointed out by other reviewers. I consider this paper suitable to be published in this Journal.

Reply: Thank you for your hard and nice work for our manuscript. Your suggestions make our work more precise and rigorous. Thank you again for your approval.